# Mapping liquid water content in snow at the millimeter scale: An intercomparison of mixed-phase optical property models using hyperspectral imaging and in situ measurements

Christopher Donahue[1], S. McKenzie Skiles[2], Kevin Hammonds[1]

[1]Department of Civil Engineering, Montana State University, Bozeman, MT, 59717, USA
[2]Department of Geography, University of Utah, Salt Lake City, UT, 84112, USA

*Correspondence to:* Christopher Donahue (christopher.donahue2@student.montana.edu)

**Abstract.** It is well understood that the distribution and quantity of liquid water in snow is relevant for snow hydrology and avalanche forecasting, yet detecting and quantifying liquid water in snow remains a challenge from the micro- to the macro-scale. Using near-infrared (NIR) spectral reflectance measurements, previous case studies have demonstrated the capability to retrieve surface liquid water content (LWC) of wet snow by leveraging shifts in the complex refractive index between ice and water. However, different models to represent mixed-phase optical properties have been proposed, including (1) internally mixed ice and water spheres, (2) internally mixed water coated ice spheres, and (3) externally mixed interstitial ice and water spheres. Here, from within a controlled laboratory environment, we determined the optimal mixed-phase optical property model for simulating wet snow reflectance using a combination of NIR hyperspectral imaging, radiative transfer simulations (DISORT), and an independent dielectric LWC measurement (SLF Snow Sensor). Maps of LWC were produced by finding the least residual between measured reflectance and simulated reflectance in spectral libraries, generated for each model with varying LWC and grain size, and assessed against the in situ LWC sensor. Our results show that the externally mixed model performed the best, retrieving LWC with an uncertainty of ~1%, while the simultaneously retrieved grain size better represented wet snow relative to the established scaled band area method. Furthermore, the LWC retrieval method was demonstrated in the field by imaging a snowpit sidewall during melt conditions and mapping LWC distribution in unprecedented detail, allowing for visualization of pooling water and flow features.

## 1. Introduction

The distribution and quantity of liquid water within a snowpack, introduced by rain and/or melt, is relevant for multiple snow related applications including snow hydrology, remote sensing, and avalanche forecasting. In terms of snow hydrology, water is an indicator of snow energy balance and snow melt timing; the change in phase from ice to water indicates that the cold content of the snowpack is depleted, and that energy balance inputs are contributing to melt (Dewalle and Rango, 2008). Rain-on-snow can accelerate this process by contributing large energy inputs into the snowpack over a short amount of time (Mazurkiewicz et al., 2008). Water at the surface will also lower snow albedo, initiating a positive feedback loop that increases absorbed solar radiation, the main driver of snowmelt (Gupta et al., 2005). For active and passive microwave remote sensing of snow, the presence of water alters microwave signatures because of the large difference in relative permittivity between liquid water and ice (i.e., dry snow). For active microwave sensors, wet snow causes characteristic changes in microwave backscatter and reduces

penetration depth (Shi and Dozier, 1992), while for passive sensors, the emissivity of the snow surface is increased (Walker and Goodison, 1993). For avalanche forecasting, the infiltration of liquid water into the snowpack impacts snow stability (Conway and Raymond, 1993). The strength of the snowpack can be increased at lower water content, where grains form well bonded clusters, but reduced at higher water content, when water flow through pore space deteriorates a significant number of snow grain bonds resulting in relatively cohesionless particles (Colbeck, 1982). Although it is recognized as a critical snow property across the cryospheric sciences, liquid water content (LWC) measurements in a snowpack are notoriously difficult to accurately quantify due to the high spatial and temporal variability of liquid water distribution.

Here, the utility of mapping LWC in situ using near-infrared hyperspectral imaging (NIR-HSI) and radiative transfer model inversion is assessed. This approach leverages the segments of the near-infrared (NIR) spectrum where the optical properties of liquid water, hereafter referred to as water, vary from those of ice. To date, wet snow has been modeled using effective spheres with a known radius, referred to as the effective grain radius ($r_e$), where the optical properties of ice and water are mixed either internally or externally. In wet snow, the arrangement of water relative to ice particles and pore space varies based on the level of saturation, which may be relevant for radiative transfer modeling. For example, water saturation below 7% (pendular regime), when water is contained in menisci held in between the ice particles (Colbeck, 1979), might be best represented using an internally mixed particle model where an ice sphere is coated in water. On the other hand, water saturation above 7% (funicular regime), when ice particles become surrounded by water as it fills the pore space, might be best represented as an internally mixed-phase sphere or externally mixed media model using interstitial ice and water spheres.

Although different mixing model representations have been proposed and demonstrated (Green et al., 2002; Hyvarinen and Lammasniemi, 1987), no study has quantitively compared the different approaches, or compared LWC retrievals to established LWC measurement methods. Without intercomparing or validation, the best approach for retrieving LWC from NIR spectral reflectance has yet to be determined. Additionally, radiative transfer approaches to retrieving $r_e$ are based on the optical properties of ice and implicitly assume dry snow, and such retrievals have not been assessed for wet snow. The main objectives of this study are threefold: (1) intercompare three wet snow reflectance models against measured LWC from a dielectric measurement instrument in a controlled laboratory environment, (2) simultaneously assess effective grain size retrieval methods and their suitability for use with wet snow, and (3) demonstrate the capability of a compact NIR hyperspectral imager to simultaneously map LWC and snow grain size at the laboratory and field scales.

## 2. Background

### 2.1 Liquid water in snow

Water infiltration through snow is a spatially and temporally complex process, controlled by water saturation level, snow microstructure, and topography. Generally, water infiltration is described by two primary mechanisms: homogenous matrix flow and heterogeneous preferential flow. Matrix flow is described as the semi-uniform vertical movement of water, while preferential flow are concentrated water pathways that follow the path of least resistance that can extend deep into the snowpack, ahead of the matrix flow (Schneebeli, 1995). Although gravitational forces

primarily drive vertical movement of water in snow, large amounts of water can be diverted horizontally due to stratigraphic layers in the snowpack, such as ice crusts or capillary barriers (i.e., fine grains over coarse grains) (Waldner et al., 2004; Webb et al., 2021; Eiriksson et al., 2013). As the snowpack becomes less stratified throughout the melt season, the general pattern transforms from preferential flow to homogenous flow (Webb et al., 2018).

## 2.2 Measurement of liquid water in snow

The complexity of water movement through snow makes observations and measurements challenging. Early observations of water flow patterns through snow were made using dye tracers (Seligman et al., 1936; Gerdel, 1954), a method which is still used today. Dye tracers provide a spatial visualization of water infiltration that has been used to study processes such as preferential flow (Schneebeli, 1995; Waldner et al., 2004) and capillary barriers (Avanzi et al., 2016). While these methods remain primarily a qualitative visualization technique, Williams et al. (2010) quantified the three-dimensional (3D) spatial distribution of meltwater within a 1 m$^3$ snowpack using dye tracers and serial-section imaging. The 3D data was binarized into dry and wet categories to quantify flow features at the centimeter scale, but LWC is not obtainable using this method.

In situ measurements of LWC in snow have traditionally been measured by centrifugal separation (Kuroda, 1954), melting calorimetry (Yosida, 1940), freezing calorimetry (Jones et al., 1983), and the dilution method (Davis et al., 1993). A more detailed summary of these methods can be found in Stein et al. (1997). Generally, these methods are difficult to perform, time consuming, and have only been occasionally used since their introduction. More commonly, LWC is measured using dielectric methods at frequencies ranging from 1 MHz to 1 GHz by leveraging the large differences in the relative permittivity ($\varepsilon_r$) between water ($\varepsilon_r \approx 88$), ice ($\varepsilon_r \approx 3.15$), and air ($\varepsilon_r \approx 1$) (Tiuri et al., 1984). This is done by time domain reflectometry (TDR) or with capacitance sensors which measure the relative permittivity of snow (Lundberg, 1997; Denoth et al., 1984). Although the measured relative permittivity is primarily a function of LWC, snow density also has some influence. Therefore, dielectric methods also require a separate density measurement. Examples of currently available dielectric instruments include the Snow Fork (Sihvola and Tiuri, 1986), Denoth Meter (Denoth, 1994), A2 Photonics WISe Sensor (A2 Photonic Sensors, 2019), and the SLF Snow Sensor (FPGA Company, 2018), which was used in this study. Although these instruments make measurements quicker relative to traditional methods, they often require destructive sampling, and only provide a discrete volume-averaged point measurement. Therefore, there is currently no in situ method to effectively quantify spatial variability of LWC at a high (sub-cm) spatial resolution, which could be used, for example, to validate process-based modeling of wet snow (Hirashima et al., 2019) or initialization and validation of microwave radiative transfer models (e.g., (Wiesmann and Mätzler, 1999; Picard et al., 2013)).

Previous non-destructive measurements of LWC in snow have been made using remote sensing techniques. Like dielectric sensors, active and passive microwave sensors leverage the difference in relative permittivity between water, ice, and air. At the ground-based scale, upward-looking ground penetrating radar (upGPR) has been used to measure the volumetric LWC directly above antennas buried below a snowpack (Schmid et al., 2014). At the spaceborne scale, active and passive microwave sensors have been used to make classification maps of wet or dry snow at spatial resolutions on the order of tens of meters (e.g., (Lund et al., 2020; Walker and Goodison, 1993)).

Similarly, in the optical wavelengths, the shift in absorption patterns of ice and water across the NIR have been leveraged to map surface LWC (Green et al., 2002), which is the primary method of interest in this work.

**2.3 Modeling wet snow near-infrared reflectance**

Absorption in the optical wavelengths is described by the imaginary part of the complex refractive index. Although the absorption patterns across the NIR are similar between ice and water, there are shifts that distinguish the different phases. The spectral complex refractive index for ice (Warren and Brandt, 2008) and water at 0 ºC (Rowe et al., 2020) across NIR wavelengths is shown in Figure 1. Compared to the difference in the relative dielectric properties across the radio and microwave wavelengths, the shifts in the imaginary part of the complex refractive

index are relatively minor, and therefore require measured reflectance at multiple wavelengths and radiative transfer modeling to detect. Additionally, the penetration of light in the NIR wavelengths is relatively shallow, limiting detection of water to the snow surface (~2 cm). This has limited the use of optical methods to detecting surface water using either in situ spectrometer measurements, or airborne imaging spectrometer measurements (Green et al., 2002; Hyvarinen and Lammasniemi, 1987).

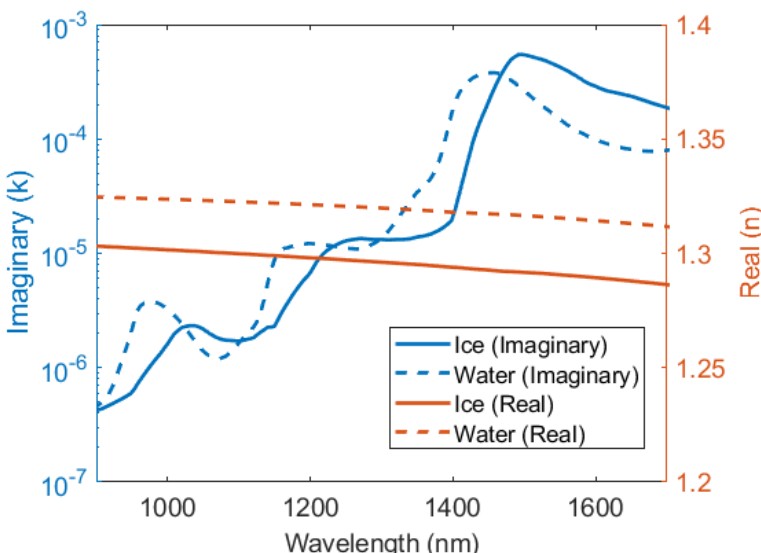

**Figure 1: Complex refractive index of ice and liquid water at 0 ºC across the near-infrared region ranging from 900 – 1700 nm, which is the same range measured by the Resonon Pika NIR-320 hyperspectral imager.**

Inversion of radiative transfer models is commonly used in remote sensing applications to retrieve physical snow properties from measured spectra and many modeling approaches have been proposed. Hyvarinen and Lammasniemi (1987) modeled the reflectance of wet snow using a collection of spheres with radius $r_e$, to simultaneously estimate LWC and grain size. To describe the optical properties of the effective spheres, an effective complex refractive index ($k_{eff}$) was calculated by volume-mixing the complex refractive index of ice and water.

Using a forward modeling approach, LWC and $r_e$ were retrieved using only three bands (1030, 1260 and 1370 nm), which were assessed using a dilatometer in a laboratory. Alternatively, Green et al. (2002) modeled wet snow using two approaches: 1) as a collection of water coated ice spheres and 2) water spheres interspersed in the interstitial space within an ice-sphere matrix. LWC and $r_e$ were retrieved by matching the simulated spectra to measured

spectra by finding the lowest residual. By visual inspection, Green et al. (2002) concluded that wet snow is best

modeled as water coated ice spheres, though no quantitative retrieval assessment was performed. More recently, a
three band ratio method to classify wet or dry snow was proposed by Shekhar et al. (2019), based on a correlation
between field spectrometer measurements and Snow Fork LWC measurements, although snow grain size was not
considered.

To date, three radiative transfer approaches for simulating the reflectance of wet snow have been proposed:

(1) mixed phase spheres, hereafter referred to as "$k_{eff}$ spheres", (2) water coated ice spheres, hereafter referred to as
"coated spheres", and (3) externally mixed interstitial ice and water spheres, hereafter referred to as "interstitial
spheres". A schematic of each model is presented in Figure 2. The $k_{eff}$ and coated sphere models are referred to as
internally mixed particles because the optical properties are mixed inside of a single particle having a single $r_e$. The
interstitial sphere model is referred to as an external mixture because the components are assumed to be physically

separate from one another.

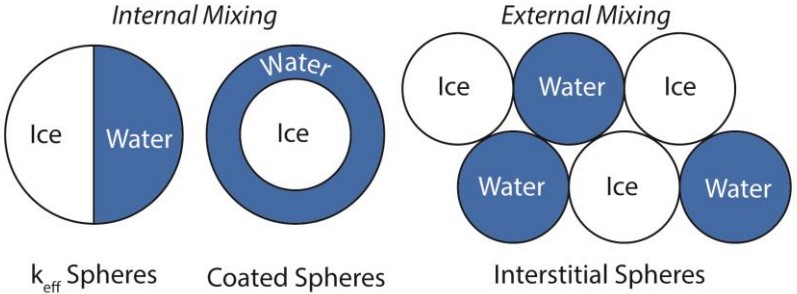

**Figure 2: A schematic of three ice and water optical mixing models used to simulate the reflectance of wet snow.**

### 3.   Methodology

Three optical property mixing models were used to simulate the bidirectional reflectance of wet snow across a range
of $r_e$ and LWC using radiative transfer modeling. To determine the optimal mixing model for retrieving LWC from
NIR reflectance, snow samples were prepared in a controlled laboratory environment with varying grain type, grain

size, and density, and then subjected to warm air advection to induce melt. During melt, time series NIR-HSI
measurements were taken and LWC was retrieved in each pixel by best matching the measured spectra to the
simulated spectra from each of the three models. For comparison to the NIR-HSI LWC retrievals, time series LWC
measurements were taken with the SLF Snow Sensor, a dielectric measurement instrument, and compared to the
average LWC from pixels covering the same measurement area as the SLF Snow Sensor, such that the two

measurements would be comparable on the same spatial scale. Time series measurements of $r_e$ were retrieved
simultaneously with LWC and were compared against the established scaled band area method of Nolin and Dozier
(2000), which has been previously applied to the same compact hyperspectral imager used in this study to map the $r_e$
of dry snow (Donahue et al., 2021). Lastly, retrievals were demonstrated in the field across an image of a snowpit
wall, visualizing water infiltration and quantifying vertical LWC and $r_e$ distributions. The modeling and in situ

measurements are described in more detail in sections 3.1-3.3, and the retrieval and assessment workflow is
represented in Figure 3.

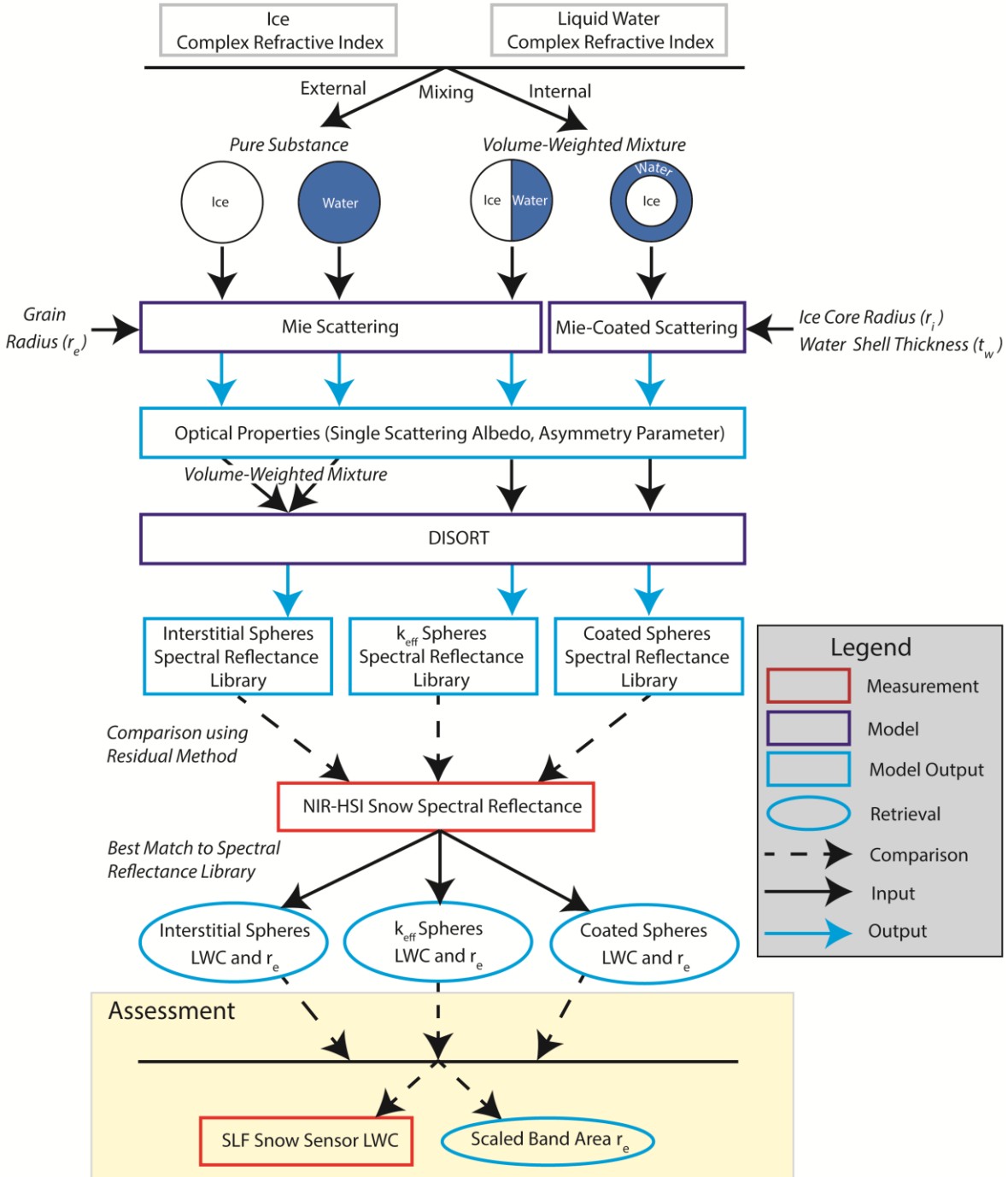

**Figure 3: Liquid water content and effective grain radius retrieval and assessment workflow.**

### 3.1 Instruments

### 3.1.1 Near-infrared hyperspectral imager

Snow reflectance in the NIR was mapped with a Resonon Inc. Pika NIR-320 near-infrared hyperspectral imager. A
brief description of the instrument follows, for a more detailed description see Donahue et al. (2021). The imager

has a spectral resolution of 4.9 nm, measuring 164 bands across the NIR region from 900 -1700 nm. The imager constructs a 2-D image containing the full spectrum in each pixel by collecting the image line-by-line, known commonly as a "push-broom" or "line-" scanner. Thus, to collect an image, the camera needs to be moving (translating or rotating) relative to the scene, or the scene needs to be moving relative to the imager. Here, both types of image acquisition techniques are used. In the laboratory, a linear scanning stage was used to move the sample beneath the sensor, while in the field, a rotational stage mounted on top of a tripod was used to scan the snowpit wall.

### 3.1.2 SLF Snow Sensor

The SLF Snow Sensor (FPGA Company, 2018), hereafter referred to as the "SLF sensor", is a capacitance sensor that is placed on the snow surface to measure the relative permittivity. This is used to determine snow density and LWC, in dry snow and wet snow conditions, respectively. The factory calibration for the LWC measurement is based on an empirical equation derived from reference measurements of snow with varying wetness and density using the dilution method (Davis et al., 1985) and weighted volumes. The sensor measures a snow surface area of 45 x 95 mm and the penetration of the electric field into the snow is ~17 mm. The SLF sensor produces a spatially comparable measurement to the retrieved LWC from NIR-HSI method presented here because the penetration of NIR light into snow is similarly shallow. Additionally, time series LWC measurements over the same surface area can be made because the sensor is non-destructive to the snow surface.

### 3.2 Experimental setup

### 3.2.1 Laboratory

The hyperspectral imager was mounted onto the Resonon Benchtop Linear Scanning Stage, which positions the imager on a stationary tower above a linear translating stage where samples are placed, shown in Figure 4a. The lens of the imager is surrounded by four halogen lamps and both are positioned for nadir viewing and illumination. The halogen lamps and lens of the imager were at a height of 38 cm and 47 cm above the snow surface, respectively. This quasi-monostatic configuration, results in a bidirectional reflectance measurement in each pixel of the image when calibrated using a white reference panel. A large Spectrolon© 99% reflectance panel was placed at the same height as the surface of the snow samples and filled the imagers entire field of view, such that each snow sample was calibrated from radiance to reflectance on a pixel-by-pixel basis. This method of calibration is ideal for hyperspectral imaging because it minimizes effects to illumination imperfections across the scene.

Snow samples were prepared in the laboratory using laboratory-made and collected natural snow to generate a dataset with a range of grain types including precipitation particles (PP), decomposing and fragmented precipitation particles (DF), rounded grains (RG), melt forms (MF), and faceted crystals (FC) (Fierz, 2009). The dry snow density, measured by weighing the sample container with a known volume and using the SLF sensor, ranged between 115 and 510 $\frac{kg}{m^3}$. Detailed properties for each snow sample are given in Table 1. Snow samples were made by sieving snow into a rectangular wooden sample container having dimensions of 17.1 cm x 12.4 cm x 8.5 cm (H x W x D). The wooden sample container was subdivided into three regions of interest (ROI), each having dimensions slightly larger than the SLF sensor, such that LWC measurements would not be impacted by edge effects from the

sample container. The surface of each snow sample was scraped with a crystal card to create a flat surface level with the top of the sample container and to minimize surface roughness. Snow samples were then kept in a cold room at -10 °C for 24 hours to equilibrate and ensure the sample was completely dry (i.e., LWC = 0%).

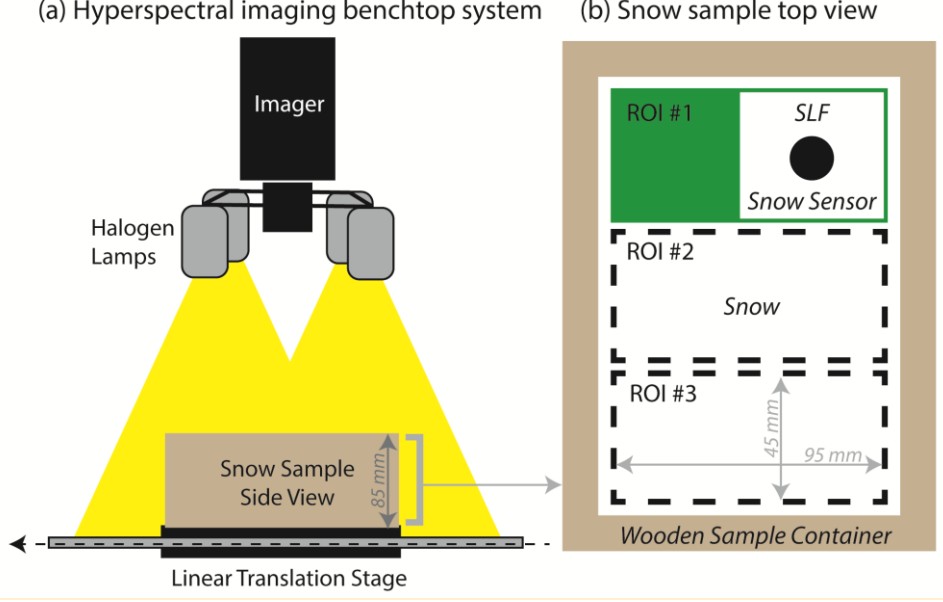

**Figure 4: Schematic of laboratory setup. (a) Front view of the Resonon Hyperspectral Imaging Benchtop System and wooden snow sample container. (b) Top view of the snow sample container showing the three regions of interests (ROI) measured by the SLF Snow Sensor, which is shown inside of ROI #1.**

For each snow sample, an initial image was taken while the cold room was at -10 °C to obtain the dry snow condition. The dry snow surface density in each ROI was measured using the SLF sensor and used as the input dry snow density to the SLF sensor for the subsequent LWC measurements. Following dry snow measurements, the cold room was turned off and the door was opened to ambient air, gradually increasing the air temperature in the cold room to room temperature (20 °C). The NIR-HSI images were taken every 1-3 minutes during the warming process

and SLF sensor measurements, in each ROI, were taken in between images. For comparison between the two

Table 1: Laboratory snow samples (reported $r_e$ is from scaled band area method).

| Sample Number | Description | Sieve Size (mm) | Initial $r_e$ (μm) | Dry Density (kg m⁻³) | Warming Time (min) |
|---|---|---|---|---|---|
| 1 | PP, Snowmaker Snow | 2 | 113 | 115 | 111 |
| 2 | DF, Snowmaker Snow | 2 | 130 | 212 | 161 |
| 3 | RG, Natural Snow | 2 | 176 | 455 | 118 |
| 4 | MF, Natural Snow | 5 | 398 | 440 | 103 |
| 5 | FC, Natural Snow | 2.5 | 463 | 493 | 86 |
| 6 | MF, Sample 4 Melt/Refreeze 1x | N/A | 699 | 468 | 99 |
| 7 | MF, Sample 4 Melt/Refreeze 2x | N/A | 898 | 510 | 208 |

instruments, the mean LWC was calculated over the 10,717 pixels within each ROI, resulting in a 0.4 mm$^2$ resolution, and was compared against the corresponding SLF sensor measurements at each time step, creating a densely populated comparison dataset spanning a wide range of LWC.

### 3.2.2 Field

To demonstrate the applicability of the NIR-HSI method for retrieving LWC and $r_e$ in the field, natural snow was imaged across the vertical wall of a snowpit. The snowpit was excavated to the ground within a protected study plot adjacent to the Alpine Weather Station, at Bridger Bowl Ski Area (Bozeman, MT; 45.82902 N, -110.92227 W) on 03 April 2021. The total snow depth was 150 cm, the snowpit wall was 143 cm wide, and there was approximately 2 meters of working room behind the snowpit wall. The day was selected because the two days preceding were sunny

and diurnal temperatures did not drop below freezing, making the likelihood of imaging wet snow high. Before imaging, standard snowpit observations were collected including a temperature profile, snow density profile using a 1000 cm$^3$ wedge cutter, and stratigraphy with grain types.

The imager was mounted on to the Resonon Outdoor Field System, which includes a tripod mounted rotational stage, and was placed 110 cm from the snowpit wall. The snowpit wall was illuminated with two 500-watt

halogen lamps mounted on a tripod, line powered (120V AC) through the weather station. The lights were placed perpendicular to the wall at a distance of 90 cm, similar to the laboratory setup presented in Donahue et al. (2021). For controlled lighting conditions, sun light (direct and diffuse) was blocked by placing an opaque tarp over the top of the snowpit. A detailed schematic of the field setup is shown in Figure 5. For a pixel-by-pixel calibration of the NIR-HSI measurements from radiance to reflectance, a 36% spectrally flat reflectance calibration tarp was hung in

front of the snowpit wall, completely covering the field of view of the imager and ROI of the snowpit.

Images of the wall were taken at 13:00, at which time there were few clouds, and the air temperature was 10 ºC. Prior to imaging the snowpit sidewall, the entire face of the snowpit was cut back ~10 cm to minimize impacts from exposure to ambient air temperature. The entire vertical profile of the snowpit was not captured in a single image, therefore two images were taken and stitched together capturing only the upper 110 cm of the snowpit

at a 6.5 mm$^2$ pixel resolution. The bottom 40 cm of the snowpit was not captured because perpendicular illumination conditions could not be achieved with the minimum height of the lighting and imager tripods used.

Immediately following imaging, LWC measurements were made with the SLF sensor along a vertical profile at 5 cm increments. For optimal LWC measurements, the SLF sensor requires the dry snow density, however, the snow was already wet and therefore the dry snow density could not be obtained. Instead, the density of

the snow from the adjacent density cut measurement was used, which introduced a small error in the LWC measurement. Following the methodology proposed by FPGA Company (2018), this error was corrected by subtracting the mass of water from the wet snow density based upon the initial LWC. The updated density was used to calculate an updated LWC using the empirical calibration equation. This calculation was repeated, with each iteration returning a smaller change in snow density. Through multiple iterations it converges on the dry snow

density and corrected LWC.

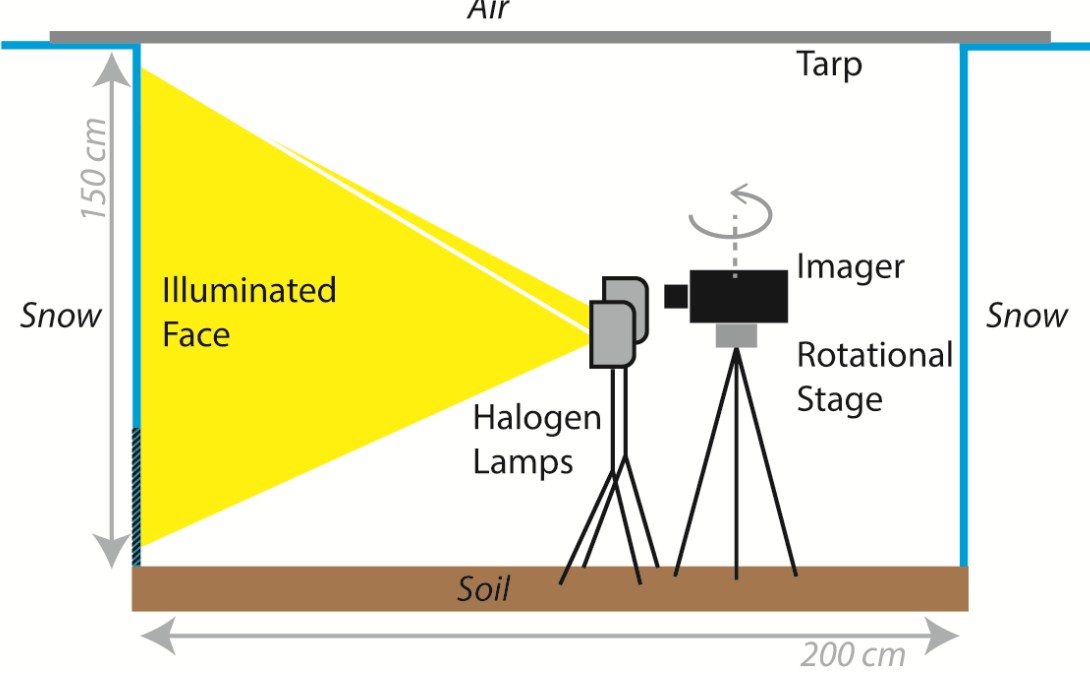

**Figure 5: Schematic of the near-infrared hyperspectral imaging setup in the field for measurement of liquid water content across a snowpit wall. The area at the bottom 40 cm of the snowpit was not imaged and is shown with hatched lines.**

### 3.3 Radiative transfer modeling

### 3.3.1 Single scattering

To simulate the optical properties of snow, the single scattering optical properties of constituents (ice, air, water, and impurities), as well as their relative arrangement to one another must be represented. The scattering properties of a single particle are described using three dimensionless optical parameters: (1) the absorption efficiency $Q_{abs}$, (2) scattering efficiency $Q_{sca}$, and (3) asymmetry factor $g$ (Bohren and Huffman, 2008). Dry snow particles are often assumed to scatter as a collection of ice spheres with radius $r_e$ (e.g., Nolin and Dozier (2000)). With a known $r_e$ and complex refractive index, Mie scattering theory (Bohren and Huffman, 2008) can be used to calculate the three dimensionless optical parameters. For clean dry snow, this modeling approach is straightforward because only the complex refractive index of ice is needed. For wet snow, on the other hand, the complex refractive index of ice and water is volume-mixed, and there are multiple approaches that can be used to define the arrangement of ice and water relative to each other. The three previously proposed mixing models used in this comparison, defined in Sect. 2, are described here.

First, the $k_{eff}$ sphere model uses a collection of spheres with $r_e$ and $k_{eff}$, which was determined using volume-weighted portions of the complex refractive index of ice ($k_{ice}$) and water ($k_{water}$).

$$k_{eff} = (1 - \%LWC) * k_{ice} + \%LWC * k_{water} \tag{1}$$

Second, the coated sphere model used the Mie-Coated scattering code of Mätzler (2002). The radius of the ice core ($r_i$) and thickness of the water coating ($t_w$) was determined by volume-weights of ice ($V_i$) and water ($V_w$).

$$V_i = \frac{4}{3}\pi r_i^3 \tag{2}$$

$$V_w = \frac{\%LWC * V_{ice}}{1 - \%LWC} \tag{3}$$

$$t_w = \left[\frac{3 * V_{water}}{4\pi + r_i}\right]^{\frac{1}{3}} - r_i \tag{4}$$

$$r_e = r_i + t_w \tag{5}$$

Lastly, the interstitial sphere model calculates the single scattering optical properties (i.e., $Q_{abs}$, $Q_{sca}$, and $g$) of pure ice spheres and water spheres separately, each having the same $r_e$. Then, the optical properties were mixed using a volume-weighted average. This method is similar to the $k_{eff}$ spheres model; however, the differences are a result of the non-linearity of Mie scattering theory (Lesins et al., 2002). At shorter wavelengths, the differences in reflectance are small, but at larger wavelengths a notable divergence occurs, shown in Figure 6.

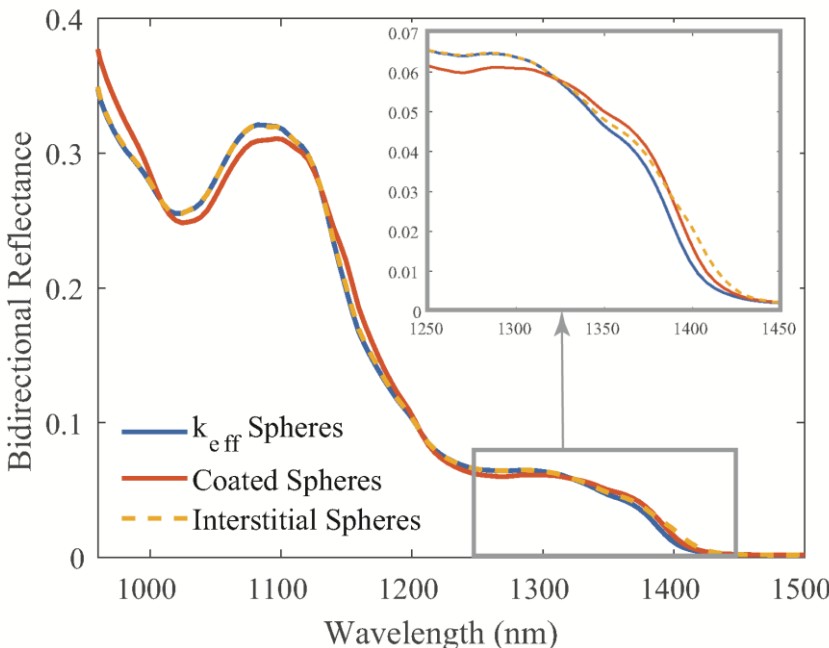

**Figure 6: Simulated bidirectional reflectance of snow using three optical mixing models: (1) $k_{eff}$ spheres, (2) coated spheres, and (3) interstitial spheres. For comparison, each mixing model has an effective grain radius of 1000 μm and 20% liquid water content by volume.**

For each model clean snow is assumed and the effects of light absorbing particles (LAPs i.e., dust and soot) are not considered, the effects of which are discussed further in section 5.3.1 The single scattering optical properties were calculated for $r_e$ values ranging from 30-1500 μm in 10 μm intervals and 0 to 25% LWC in 1% intervals,

resulting in 3,848 simulated combinations per model. Single scattering optical properties for each case were then used as inputs to solve for snow reflectance, the result of multiple scattering events.

### 3.3.2 Multiple scattering

To generate a spectral library to match to measured spectra, directional-hemispherical reflectance for each mixing model was simulated using a general-purpose 16-stream plane-parallel discrete ordinates radiative transfer model, DISORT (Stamnes et al., 1988). DISORT allows the user to define optical properties of multiple layers; here, a single optically thick layer was used since the penetration of NIR light into the snowpack is shallow. Optical property inputs for this layer included single scattering albedo, defined as the ratio of the scattering efficiency and

extinction efficiency $\left(\frac{Q_{sca}}{Q_{sca}+Q_{abs}}\right)$, and $g$, which were computed from Mie scattering theory. Incoming light can be modeled at multiple user defined zenith angles; here, the zenith angle was set to $0^\circ$ to represent nadir lighting.

The output from DISORT was directional-hemispherical reflectance, whereas NIR-HSI measurements are bidirectional reflectance. This is a suitable approach because of the experimental setup: the NIR-HSI measurements were made at nadir illumination and viewing angles and calibrated using a Lambertian white reference target. Under

these conditions, snow is nearly Lambertian, allowing for a direction comparison, which would not be the case for non-nadir viewing angles, given that snow heavily favors forward scattering (Dumont et al., 2010).

### 3.3.3 Retrieving snow properties

To simultaneously retrieve LWC and $r_e$, measured spectra in each pixel of the NIR-HSI image was compared to each spectrum in the spectral library to determine the best match by finding the minimum least square residual across 106

bands, ranging from 961 nm to 1472 nm. Hereafter, this method for matching measured and simulated spectra is referred to as the "residual method". An example of (1) measured spectrum from NIR-HSI, (2) retrieved simulated spectra using the interstitial sphere model, and (3) residuals across each band are shown in Figure 7. Once the measured spectrum is best matched to a simulated spectrum, the associated $r_e$ and LWC are assigned to the pixel, producing separate maps of $r_e$ and LWC for the imaged area. To reduce impact from sensor noise at the lower limit

of the sensor (900 nm), 961 nm was chosen as the starting point for calculating the residuals while still capturing ample spectral data for the left side of the absorption feature centered at 1030 nm (Figure 7). At longer wavelengths, 1472 nm was chosen as the endpoint for calculating residuals because both ice and water are highly absorptive beyond this point, resulting in a low signal to noise ratio.

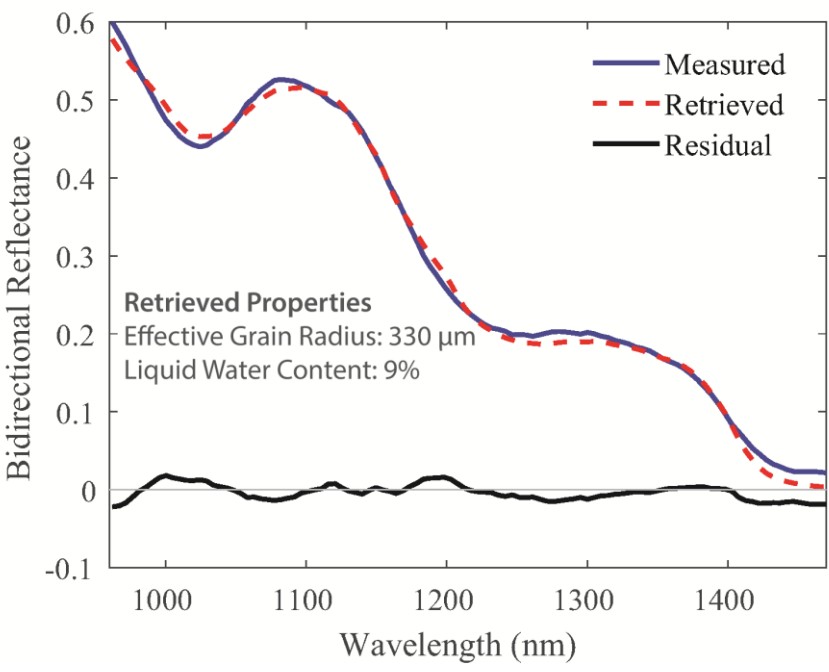

**Figure 7: Example of a NIR-HSI measured spectrum, retrieved simulated spectrum using the interstitial sphere model, and the residuals at each band.**

Additionally, $r_e$ was mapped at each timestep using the scaled band area method following Donahue et al.
(2021). Briefly, the scaled band area is the area underneath a continuum line spanning an absorption feature and is a shape-based method that is independent of absolute reflectance. Here, the scaled band area was calculated for measured spectra using a predefined start and end point for the continuum line across the absorption feature centered at 1030 nm. The pixel-by-pixel calibration performed here reduced noise at the lower limit of the sensor compared to Donahue et al. (2021), allowing for the defined continuum endpoints to be similar to bands suggested by Nolin and Dozier (2000) (i.e. 961 and 1087 nm). A look up table was populated with "dry snow" (all ice, no water) scaled band areas for simulated $r_e$ ranging from 30-1500 μm. We note that we used the simulated spectra from the interstitial sphere model, but that the dry snow representations for all mixing models were identical, with variation in spectra introduced only when water was represented. This allowed us to (1) define the starting $r_e$ for each of the prepared laboratory samples (Table 1), (2) compare $r_e$ retrieved from scaled band area to that from the residual method, and (3) assess the suitability of the scaled band area method for grain size retrievals over wet snow.

## 4. Results

### 4.1 Laboratory experiments

### 4.1.1 Liquid water content retrieval

The LWC retrieved from NIR-HSI was compared to the SLF sensor across 7 samples, spanning a wide range of initial dry snow grain sizes from approximately 100 to 900 μm, measured using the established scaled band area method. LWC measured with the SLF sensor ranged from 0 to 17% across 21 ROIs and 40 timesteps, producing 690 datapoints for comparison. It was found that the interstitial sphere model consistently performed the best, whereas the coated sphere model performed the most poorly, relative to the in situ SLF sensor measurements.

An example of the LWC maps produced as melt progressed in a single snow sample is presented in Figure 8. These examples are of the same ROI and show LWC retrieved using the interstitial sphere model. The initial image (8a) was taken at the start of the experiment when the snow was dry and 98% of pixels (10,450 pixels) retrieved 0% LWC. The other 2% of pixels (267 pixels) retrieved 1% LWC, which was found to be due to sensor noise. The remaining images in Figure 8 capture melt progressing through 5% (b), 10% (c), and 16% (d) mean LWC and the corresponding SLF sensor measurement is noted in each panel. Additionally, the distribution of values also broadens with increasing LWC, shown in the per-pixel distribution of LWC for each image in Figure 8e. The summary statistics show melt progression, as expected, but the maps allow visualization and quantification of melt initiation and LWC distribution. The melt features that begin to develop in early time steps can be tracked to later time steps (e.g., 8c to 8d).

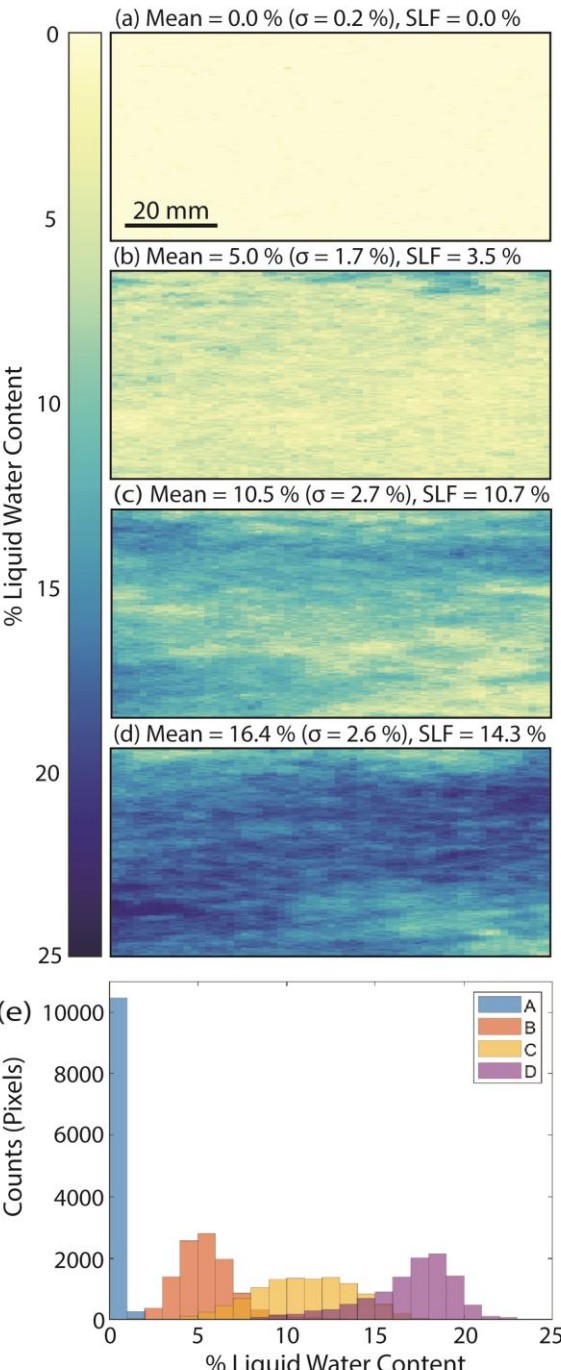

**Figure 8: (a-d) Time series liquid water content mapping over a single region of interest during a laboratory experiment. (e) Liquid water content distribution in images (A-D) shown in a histogram.**

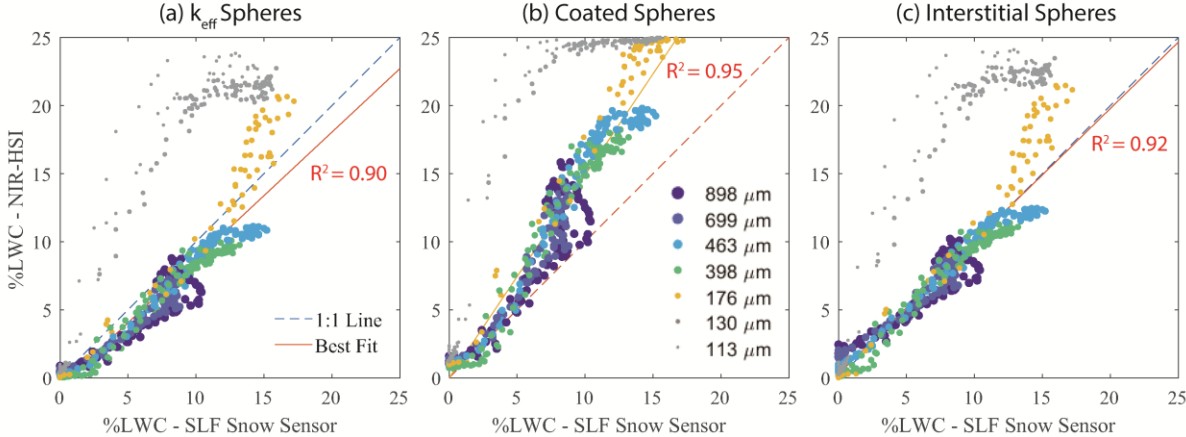

**Figure 9: Comparison of LWC measured with the SLF Snow Sensor versus the mean LWC retrieved from NIR-HSI using the residual method. Three optical property mixing models are compared using the same datasets: (a) $k_{eff}$ spheres, (b) coated spheres, and (c) interstitial spheres. The initial (dry) effective grain radius for each sample is shown in the legend and the $R^2$ value is for the best fit line, which excludes 113 and 130 μm datasets.**

The full performance comparison across all datasets is summarized in Figure 9, which plots LWC from the SLF sensor against that from each mixing model applied across all samples. To help visualize the difference between experiments, the comparison points are symbolized by different colors as well as marker size that varies with the mean initial dry snow $r_e$ retrieved using the scaled band area method. Close proximity to the 1:1 line would indicate the best match between NIR-HSI and the SLF sensor. The root mean square error (RMSE) and bias for each sample is summarized in Table 2.

**Table 2: Liquid water content retrieval results from the laboratory (reported $r_e$ is from scaled band area method).**

| Sample Number | n | Initial $r_e$ (μm) | %LWC Range | RMSE (% LWC) | | | Bias (% LWC) | | |
|---|---|---|---|---|---|---|---|---|---|
| | | | | $k_{eff}$ Spheres | Coated Spheres | Interstitial Spheres | $k_{eff}$ Spheres | Coated Spheres | Interstitial Spheres |
| 1 | 66 | 113 | 0 - 14.6 | **8.7** | 11.9 | 9.4 | **7.5** | 10.5 | 8.0 |
| 2 | 102 | 130 | 0 - 15.9 | **7.7** | 11.5 | 8.6 | **7.1** | 10.8 | 7.9 |
| 3 | 78 | 176 | 0 - 17.2 | **2.0** | 7.0 | 2.6 | **1.0** | 5.8 | 1.6 |
| 4 | 123 | 398 | 0 - 13.4 | 1.8 | 4.0 | **0.9** | -1.5 | 3.2 | **-0.5** |
| 5 | 132 | 463 | 0 - 15.2 | 1.9 | 4.7 | **1.1** | -1.5 | 4.0 | **-0.3** |
| 6 | 102 | 699 | 0 - 8.8 | 1.6 | 2.8 | **1.0** | -1.2 | 2.3 | **0.3** |
| 7 | 90 | 898 | 0 - 10.4 | 2.2 | 3.3 | **1.4** | -1.6 | 2.3 | **-0.1** |

For the two samples with the smallest initial $r_e$ (113 and 130 μm), precipitation particles and decomposed/fragmented particles, it was found that the LWC retrieval method did not perform well using any of the mixing models, with the highest RMSE and bias (Table 2). The retrieval is near 0% LWC in dry snow conditions, however the LWC retrievals increase rapidly as small amounts of water are introduced. In the coated spheres model, the retrievals reach the LWC limit of the spectral library (25%) when the measured LWC from the SLF sensor was ~10% (Figure 9b). For $k_{eff}$ and interstitial spheres, there is a similar pattern of LWC increasing too fast, although these models do not reach the upper limit of the spectral library. Because none of the models evaluated performed

well for small grain sizes, these two samples (113 & 130 μm) were excluded from the calculation of best fit line (red line in Figure 9). This result is discussed further in Sect. 5; however, the exclusion of these data is reasonable because water is not commonly mixed with these types of particles (new snow) in natural environments and therefore not a primary focus for wet snow mapping applications.

For the remaining samples, ranging from 176 – 898 μm, the mixing models retrieve LWC values that more closely match the SLF sensor measurements. Visually, both $k_{eff}$ and interstitial spheres fall close to the 1:1 line. The $k_{eff}$ spheres do have the lowest RMSE and bias for the smallest grains (samples 1-3), but for the remaining medium to large grain size samples, the retrieved LWC consistently has a negative bias and the RMSE is ~2%. Overall, the interstitial spheres have the lowest RMSE (~1%) and bias, which does not trend with grain size, with the best comparison for samples 4-7 and similar values to $k_{eff}$ for sample 3. The uncertainty of the interstitial sphere model was determined by taking the mean RMSE across samples 3-7, which is 1.4%. This is supported by the dry snow retrieval shown in Figure 8A, where no pixels retrieved greater than 1% LWC. The coated spheres model performed most poorly relative to measurements with a high RMSE and consistently had a positive bias across all samples, although like the other mixing models, the values do fall close to the 1:1 line at low LWC (< 7%). Overall, these experiments and model comparisons show that the interstitial spheres model performed exceptionally well for medium to large grains, and LWC ranges between 0% and 15%, which are the conditions most likely to be found in natural snow covers.

### 4.1.2 Effective grain radius retrieval

Using the residual method, all mixing models retrieved similar grain size values because the grain size retrievals are primarily dependent on the absolute reflectance driven by ice absorption. Here, we present results from the interstitial sphere model because it performed best in the LWC retrieval. For initial dry snow conditions, the $r_e$ retrieval using the scaled band area method ($r_{e, SBA}$) had a positive bias relative to the residual method ($r_{e, residual}$), and the bias becomes more positive with increasing grain size (Figure 10). For the remaining wet snow comparisons, the $r_{e, SBA}$ remains relatively constant or increases at low LWC followed by a decrease at high LWC. For Samples 1 and 2, $r_{e, SBA}$ remained flat with increasing LWC. For Samples 3-6, $r_{e, SBA}$ increased initially with low LWC and then decreases at high LWC. For Sample 7, $r_{e, SBA}$ slightly increases before significantly decreasing with increasing LWC. Whereas the $r_{e, residual}$ increases with increasing LWC for all samples, which is expected because the presence of water is known to accelerate snow grain growth (Marsh, 1987). The comparison, further discussed in Sect. 5.2, shows that the scaled band area method is impacted by the presence of water, such that the residual method may be better suited for wet snow.

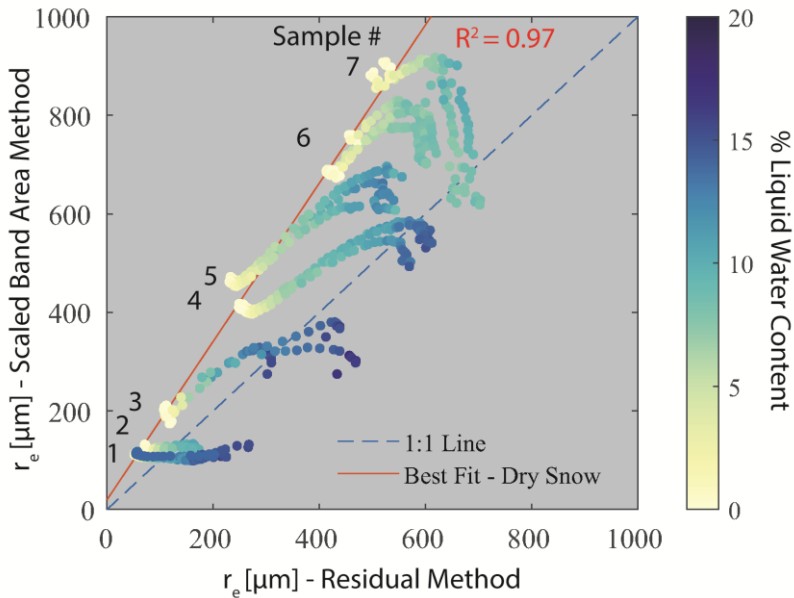

**Figure 10:** Mean effective grain radius ($r_e$) retrieval comparison between residual and scaled band area methods using the interstitial sphere model at each time step for Samples 1-7. Best fit line is drawn through the dry snow condition for each sample.

### 4.2 Field experiment

Although a controlled laboratory environment is ideal for identifying the best suited optical mixing model, the primary applications for this method would be in situ field studies, motivating the field-based testing of the $r_e$ and LWC retrieval. The maps of $r_e$ and LWC generally reflect the profile measurements of LWC and stratigraphy, though at significantly higher detail (Figure 11). The snowpit was representative of a spring intermountain snowpack undergoing melt, with the density values ranging from ~300 to ~450 kg m$^{-3}$, and LWC measurements ranging from ~6-16%. The temperature profile, not shown, was isothermal at 0 °C. The general stratigraphy was ice lenses and melt form grains in the upper layers, rounded grains in the central portion, and faceted grains (depth hoar) near the ground. Note that the full pit profile is shown for observations (Figure 11a and 11d), while the $r_e$ and LWC retrievals extend across only the upper 110 cm (Figure 11b and 11c).

The maps were processed using each of the mixing models introduced above, but based on the laboratory findings, only retrievals from the interstitial sphere model are presented here. The SLF sensor measurements were taken along the left side of the ruler (gray stripe or NaNs down the center of snowpit), represented as the dashed red box in Figure 11c. For comparison, LWC was depth averaged in pixels covering the area of the SLF sensor measurements (red line in Figure 11d), in addition to the depth average LWC across the entire width of the snowpit (gray line in Figure 11d). The mean LWC, standard deviation ($\sigma$), and number of measurements (n) for the SLF sensor, NIR-HSI along the same profile as the SLF sensor, and the depth average of the entire width of the snowpit are shown in Table 3. Generally, the LWC from the NIR-HSI retrieval had a positive bias compared to the SLF

400    sensor, particularly notable at the top of the snowpack, although the patterns and peaks have similar trends. The

difference between retrievals and measurements is discussed further in section 5.1.

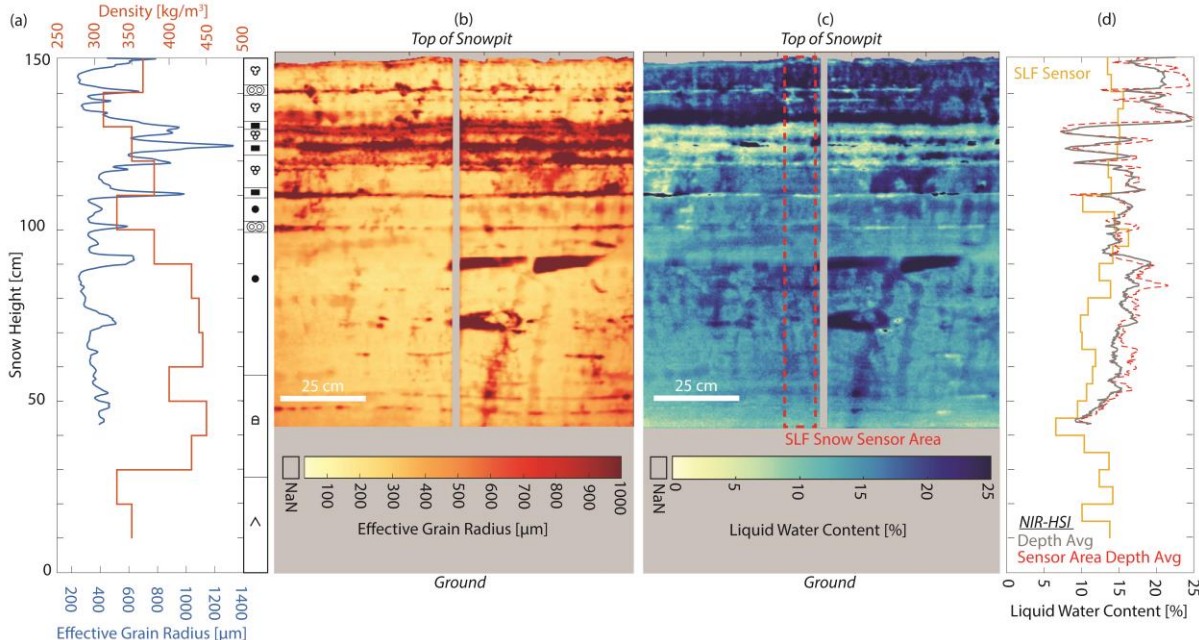

**Figure 11: Results from the snowpit at Bridger Bowl Ski Area. (a) Density profile using 1000 cm³ wedge cutter and depth averaged effective grain radius. (b) Map of effective grain radius across snowpit wall. (c) Map of liquid water content across snowpit wall with the SLF sensor vertical profile area outlined in dashed red line (d) Liquid water content profiles from the SLF Snow Sensor, NIR-HSI depth averaged across the entire width of the snowpit, and NIR-HSI depth averaged over the area covered by the SLF sensor area only. NaN (Not a Number) values in (b) and (c) correspond to no retrieval data due to a ruler that was placed in the scene.**

For comparison, when processed with the coated spheres model the LWC reached the model limit at 25%

over much of the scene. The $k_{eff}$ spheres model did have mean values slightly closer to the SLF sensor, however this

was not unexpected given that the laboratory results were biased negative. The magnitude of values, with coated

405    spheres retrieving higher values, and $k_{eff}$ spheres retrieving lower values, relative to interstitial spheres mimics the

bias present in laboratory results (Table 2).

**Table 3: Liquid water content retrieval results from the field.**

|  | **SLF Snow Sensor** | **K$_{eff}$ Spheres** | | **Coated Spheres** | | **Interstitial Spheres** | |
|---|---|---|---|---|---|---|---|
|  |  | Sensor Area | Full Profile | Sensor Area | Full Profile | Sensor Area | Full Profile |
| **Mean LWC [%]** | 12.6 | 15.6 | 15.1 | 23.5 | 23.2 | 16.3 | 15.7 |
| **σ** | 2.3 | 3.2 | 2.7 | 1.8 | 1.1 | 3.4 | 3.2 |
| **n** | 28 | 21,758 | 243,386 | 21,758 | 243,386 | 21,758 | 243,386 |

The high resolution of the maps show how stratigraphy influences $r_e$ and LWC distributions in higher detail

than can be captured with standard field-based observations. At the top of the snowpit, it can be discerned that the

snow was saturated with water, which was pooling on top of the ice lens at 130 cm snow height. The SLF sensor only captures the average LWC, particularly between 110 and 130 cm, where there is high variability in LWC between layered ice lenses. At the ice lenses (130, 122, and 110 cm) the grains are large while the LWC is low. Water pooling above ice lenses, rather than the ice lens itself having water content, is sensible because there is minimal pore space for water to reside or pass through. Below the ice lens, a few isolated preferential flow paths extend to lower layers, while other features show water concentrated (at ~90 cm, and at ~75 cm) but not flowing along the plane of the snowpit wall.

## 5. Discussion

### 5.1 Liquid water content retrieval

The interstitial and $k_{eff}$ spheres models performed similarly because the optical properties are volume-mixed in both cases, albeit internally versus externally mixed. The external mixing of interstitial water spheres results in a notable shift in the reflectance spectrum at wavelengths ranging from 1300 – 1450 nm, when compared to $k_{eff}$ spheres (Figure 6). Since the interstitial sphere model performed the best, this result indicates that the particle size of water in wet snow plays an important role in the simulated reflectance, whereas the particle size of water is not considered in the $k_{eff}$ model. Before mixing the optical properties, the particle size of the water sphere was the same size as the ice sphere, which is a reasonable approximation. It would be possible to mix water and ice spheres of differing size, although this approach is computationally expensive, given that the number of possible simulated spectra combinations would approach infinity.

The coated sphere model performed reasonably in the pendular regime, where water is contained in menisci held in between the ice particles, but then considerably overestimated LWC in the funicular regime. The coated sphere model was chosen over the interstitial sphere model by Green et al. (2002), based on visual inspection, considering the bands in the ice absorption feature centered at 1030 cm, which encompasses only part of the distinct shifts between ice and water that are present in the complex refractive index across the NIR (Figure 1). This study used a greater number of NIR bands that span multiple distinguishable shifts between ice and water, which is a more robust approach.

For small snow grains of PP (Sample 1) and DF (Sample 2) crystal type, LWC retrievals did not perform well using any of the models. One potential reason being that PP and DF crystal types are complex shapes and reflectance may not be accurately represented using spheres. Using a ray tracing model, Picard et al. (2009) showed that grain shape can influence reflectance. Although it is possible to have wet PP and DF crystal types (e.g., rain on snow), low density dendritic snow crystals are more commonly found at temperatures well below freezing (Judson and Doesken, 2000). It is far more common for wet snow to contain larger rounded grains primarily because the presence of water rapidly increases the rate of snow grain growth, especially through melt and refreeze cycles. Small RG (Sample 3) having only slightly larger $r_e$ values than the PP and DF crystals, performed similarly to the medium and large sized grains, further suggesting that the complex shapes of PP and DF may be driving the poor performance. Based on the results at small grain sizes, the mixing models have potential to classify dry versus wet snow, but not quantitively retrieve LWC.

445        Interestingly, for the largest grains, samples 6 and 7, the highest LWC measurements from the SLF sensor are ~10% (Figure 9). These apparent maxima, seen in both measurements and retrievals, is attributed to the large grains having a reduced water holding capacity within the pore space (Yamaguchi et al., 2010). Although the snow sample could contain higher than 10% LWC by volume, water is able to drain below the near surface detection limit of the SLF sensor and NIR-HSI.

**5.2 Effective grain radius retrieval**

The scaled band area method assumes dry snow, but in remote sensing and field applications there is typically no a priori knowledge of snow wetness, thus comparing $r_{e,\,SBA}$ to $r_{e,\,residual}$ allows us to test the validity of this assumption for wet snow. To visualize the difference between the residual and scaled band area methods, an example measured spectrum from a dry and wet snow (12% LWC) sample are shown in Figure 12, along with the corresponding

simulated spectrum retrieved using the two methods. For the dry snow spectrum, $r_{e,\,SBA}$ is larger than $r_{e,\,residual}$, which is a consistent trend across all samples (Figure 10 best fit line). This is attributed to the scaled band area method using a fixed wavelength range spanning the area of the absorption feature centered at 1030 nm (961 to 1087 nm), which makes it sensitive to small changes in the shape of the measured absorption feature, and location of the continuum line shoulders (shown as gray lines in Figure 12b). Whereas the residual method finds the best fit

spectrum over all NIR wavelengths ranging from 961 nm to 1472 nm. In the example shown (Figure 12b), the left shoulder of the matched simulated spectrum using the residual method is slightly below the measured spectrum, resulting in the scaled band area of the measured spectrum (2.92) being higher than that of the matched simulated spectra (2.46).

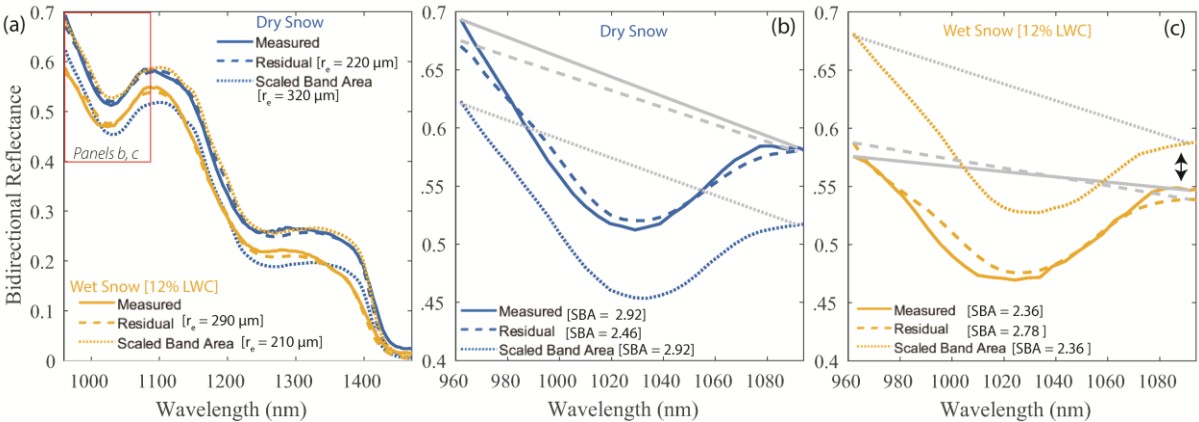

**Figure 12: Effective grain radius ($r_e$) retrieval comparison between the residual and scaled band area methods using the interstitial sphere model. (a) An example measured spectra from a dry and wet snow sample and the corresponding retreived simulated spectra using the residual and scaled band area methods. (b) Dry snow spectra from (a) including the continuum lines, shown in gray, over the absorption feature. (c) Wet snow spectra from (a) including the continuum lines over the absorption feature, shown in gray.**

       In the wet snow case, the absorption features centered at 1030 nm shifts to shorter wavelengths and

broadens. Similar to the comparison for dry snow, the fixed wavelength range and continuum line of the scaled band area method fails to fully capture the wet snow absorption feature, resulting in a reduction of the scaled band area.

This result is shown in the spectral reflectance example (Figure 12c) and is responsible for the decreasing $r_{e, SBA}$ at high LWC seen for samples 3-7 (Figure 10). In this example, the shift to the left is highlighted with the arrow pointing to the flattening of the feature that is incorrectly included by the scaled band area fixed continuum line

(shown as gray lines in Figure 12c), resulting in a smaller scaled band area. This example also shows the broadening of the feature, relative to the shape of ice absorption alone. The shape of the wet snow feature is better represented by the matched simulated spectrum using the residual method. It is possible that this has implications for previous studies that have applied the scaled band area method over potentially wet snow (e.g., Skiles and Painter (2017)).

### 5.3 Uncertainties

### 5.3.1 Simulated Snow Reflectance

All the mixing models examined in this study are approximate representations of the relative arrangement of ice and water in wet snow. The spherical particle approximation used in this study to represent wet snow is a reasonable approach because ice grains in the presence of water tend to be rounded. The arrangement of ice and water, on the other hand, is dependent on the level of water saturation, therefore using a single mixing model to determine the

LWC across a large range of water saturations results in inherent uncertainty. Since no a priori knowledge of snow wetness is known when taking NIR-HSI measurements, the goal of this research only aims to find the modeling approach that has the best retrieval of LWC across ranges commonly found in natural environments, when compared to an established dielectric method-based instrument.

Simulations of clean snow reflectance, assuming no LAPs, was valid for the laboratory experiments,

however snow in natural environments contain varying amounts of LAPs. Since LAPs predominately effect the visible part of the spectrum (Nolin and Dozier, 2000), it is unlikely that typical concentrations would impact this retrieval method. However, at high concentrations of LAPs their impact extends into the NIR, and in extreme cases this can make the ice absorption feature more shallow (Skiles et al., 2018), which would increase uncertainties in this retrieval method.

### 5.3.2 Measured Snow Reflectance

There is uncertainty in the measurement of absolute spectral reflectance, the accuracy of which is important for using the residual method. To minimize this uncertainty in the laboratory, spectral measurements were taken with consistent lighting conditions and at close proximity, resulting in near perfect conditions, which was ideal for the comparison study. Similarly, uncertainties related to lighting conditions in the field experiment were minimized by

blocking all sunlight with an opaque tarp and illuminating the snowpit wall with a known lighting source. This approach is recommended and is used with other optical methods, such as contact spectroscopy (Painter et al., 2007; Skiles and Painter, 2017). Using natural light in a standard snowpit orientation (i.e., facing away from the sun) would make the conversion from radiance to reflectance challenging because the snowpit wall is unevenly illuminated by diffuse light, and would need to be accounted for in the radiative transfer modeling. Additionally,

there may not be enough light on the snowpit wall for imaging, however, this was not tested here.

Similarly, if the orientation of the imager or light source is off-nadir, this needs to be accounted for in the radiative transfer modeling such that the measured and simulated spectra are comparable. Here, we did not image the bottom 40 cm of the snowpit because positioning the camera and lights off-nadir would introduce errors in the comparison to the simulated spectra at nadir viewing, which is discussed in more details in Donahue et al. (2021).

However, the full snowpit could be imaged in nadir orientation by digging a larger snowpit, such that the camera and lighting source are farther away from the snowpit wall, or by using a tripod that lowers closer to the ground.

Additionally, the residual method benefits from the high signal-to-noise ratio (1885) and spectral resolution (4.9 nm) of the instrument used here. For application of this method at the airborne or satellite scales, spectral measurements contain more noise than those in the laboratory and require atmospheric and topographic correction,

introducing additional uncertainty into absolute reflectance values. Although not within the scope of this study, instruments at these scales also need to account for water vapor in the air which is discussed further in Green et al. (2006). Finally, due to the relatively minor shift in NIR reflectance, this approach is likely not suitable for mixed pixels (not pure snow), which become more common as spatial coverage and pixel sizes increase.

### 5.3.3 Dielectric Liquid Water Content Measurements

Dielectric instruments, including the SLF sensor used here, have their own uncertainties. Based on the empirical calibration of the SLF sensor (FPGA Company, 2018), the RMSE of the LWC measurement is ~1.2%. These types of instruments also rely on an independent snow density measurement to calculate LWC. In the case of the SLF sensor, the "dry snow" density, which describes the fraction of ice in a wet snow volume, is required to calculate a LWC value (FPGA Company, 2018). In the laboratory experiments, this was not an issue because all snow samples

started dry, and the dry snow density was used for all subsequent LWC measurements during melt. Conversely, in the field experiment, the snow was already wet and there was no way of isolating a dry snow density. The preferred solution offered by FPGA Company (2018) is to measure density in the morning if the snow has refrozen, but this was not possible here. A second solution, described in section 3.2.2 and applied here, is to use an iterative approach to determine an estimated dry snow density. Since the measured LWC is dominated by water content rather than

snow density, the change in corrected LWC was small, ranging from 0.4-1.2%. Due to the cubic empirical calibration curve (FPGA Company, 2018), high LWC values had a smaller correction while smaller LWC values had a larger correction. Nevertheless, a known dry snow density at the time of measurement would give the most accurate LWC measurement from the SLF sensor. Although there is inherent uncertainty associated with the SLF sensor, it was found to be the most suitable LWC measurement instrument for comparison to LWC retrieved from

NIR reflectance because it is non-destructive to the snow surface, measures a flat surface, and has minimal penetration into snow, similar to NIR light.

### 5.3.4 Field Measurements

Acquiring a vertical profile of LWC, using any instrument, requires digging a snowpit and exposing the sidewall. The atmospheric exposure can change snow properties and introduce uncertainties to the measurement. Shea et al.

(2012) observed that a statistically significant change in the surface temperature across a snowpit wall occurred within the first ninety seconds of exposure, providing evidence that atmospheric equalization affects the surface

temperature more strongly than from heat behind the snowpit wall. Here, the snowpit was isothermal and the air temperature was above freezing (10 ºC) meaning that any additional energy into the snowpack directly goes to melting snow and increasing LWC. To minimize these uncertainties, a systematic approach was taken where the snowpit sidewall was cutback ~10 cm prior to taking hyperspectral images and SLF sensor measurements. Additionally, the snowpit was covered with an opaque tarp blocking all diffuse sunlight on the snowpit wall. Acquiring two hyperspectral images took approximately 2 minutes while acquiring the 28 SLF sensor measurements in a vertical profile took approximately 5 minutes, resulting in the snowpit sidewall being exposed for ~7 minutes. The comparison of the two instruments cannot occur simultaneously and requires the snowpit to be exposed for an extended time, which would lead to uncertainties if the LWC on the snowpit wall increased during this time.

### 5.4 Implications for Future Applications

Controls on liquid water movement through snow include LWC, grain size, and wet snow metamorphism (Hirashima et al., 2019). Multi-dimensional snowmelt models have been developed to represent the relationship between grain growth and water percolation (Hirashima et al., 2014), but limited observations for validation at high spatial scales currently exist. Being able to coincidently map $r_e$ and LWC spatial distributions in such high detail could support processes-based studies and validate models coupling wetting front propagation with grain size and grain growth.

Coincident $r_e$ and LWC maps could also be used to better interpret microwave remote sensing retrievals and for comparison to microwave radiative transfer models, such as the Microwave Emission Model for Layered Snowpacks (MEMLS) (Wiesmann and Mätzler, 1999) and the Dense Media Radiative Transfer Multi-Layer model (DMRT-ML) (Picard et al., 2013). Generally, these models are initialized with physical snow properties, including $r_e$, LWC, and layer thickness. Currently, discrete in situ measurements of $r_e$ and LWC measurements are taken using instruments, such as IceCube (Zuanon, 2013) and SLF sensor, respectively. These instruments are generally affordable, portable, and do not require a large snowpit working space. However, to measure snow properties multiple instruments are needed and measurements are discrete and asynchronous, making it challenging to capture spatial variability. Additionally, multiple measurements require the snowpit wall to be exposed for longer, increasing uncertainty in the measured snow properties. Although logistically more challenging to implement in the field, the NIR-HSI method addresses these limitations by measuring $r_e$, LWC, and layer thickness in high resolution simultaneously while minimizing the time the snowpit wall is exposed to ambient air.

There is also broader relevance for the assessment and development of snow property retrievals from measured spectral reflectance with upcoming satellite imaging spectrometer missions. These include the Surface Biology and Geology (SBG) imaging spectrometer mission and the Copernicus Hyperspectral Imaging Mission (CHIME). Although algorithm suites have been developed to retrieve snow properties from airborne imaging spectroscopy (Painter et al., 2013), LWC is not a standard part of the retrieval, and has only been demonstrated as a case study (Green et al., 2002). Time-series mapping of LWC could be used to monitor melt initiation and how it varies with slope, aspect, and elevation.

## 6. Conclusions

The results in this study show that the externally mixed interstitial spheres model performs best when compared to a dielectric LWC measurement instrument, relative to the previously proposed $k_{eff}$ spheres (Hyvarinen and Lammasniemi, 1987) and coated spheres model (Green et al., 2002). It was also found that for the smallest grains (i.e., new and decomposing precipitation particles) none of the models investigated compared well to the SLF sensor. For low LWC (<7%), all the retrievals compare well to measurements, but at higher LWC the $k_{eff}$ and coated spheres were biased positive and negative, respectively. Overall, and across the widest range of initial grain types, $r_e$ (162 – 859 μm) and LWC (0-17.2%), *the interstitial sphere model performed the best,* with ~1% uncertainty.

For the $r_e$ comparison, between the two optically based residual and scaled band area methods, it was found that the scaled band area retrieval had a positive bias compared to the residual method. This bias was lowest for small grains and increased with grain size. For wet snow, the scaled band area method was impacted by the presence of water due to the shift in the absorption feature to shorter wavelengths, resulting in decreasing $r_e$ at high LWC. Because scaled band area implicitly assumes dry snow, it is based on ice absorption alone, caution is encouraged when applying this method without knowing that the snow is dry.

The field application of the NIR-HSI method produced maps that reflect the general understanding of what a snapshot of a snowpit would look like during snowmelt progression, but mapped the stratigraphy of snow properties (i.e., LWC and grain size) at much higher resolution relative to standard profile-based observations. The retrieved LWC was found to be slightly higher than that measured by the SLF sensor, which was attributed to both the inability to determine the dry snow density and the high level of detail in the maps that could not be captured by the volume averaged dielectric sensor.

**Data availability**

Data will be available upon request from lead author.

**Author contribution**

CD conceptualized the study, collected laboratory and field data, conducted DISORT model simulations, and analyzed the results. MS provided the hyperspectral imager, assisted with data collection, and advised CD during conceptualization and DISORT model simulations. KH acquired funding for this research, was responsible for project administration, and supervised CD throughout the study. CD wrote the original draft manuscript and all coauthors contributed during review and editing.

**Competing interests**

The authors declare that they have no conflicts of interest.

**Acknowledgements**

This research was supported by the NASA New Investigator Program Award 80NSSC18K0822. Additionally, Skiles was supported by the NSF CZO-Net Award EAR-2012091 and NASA SBG Pathfinder Study (SISTER). We thank Joseph Shaw for providing the Resonon Linear Scanning Stage used in this study. We thank Michael Dvorsak

and Ladean McKittrick for their assistance in the laboratory. We thank Pete Maleski and the Bridger Bowl Ski Patrol for allowing us to conduct field work at the Alpine Weather Station and for transporting our instruments into the field. We acknowledge the use of the Subzero Research Laboratory in the Department of Civil Engineering at Montana State University. We thank Ryan Webb and Chander Shekhar for providing insightful comments that helped improve this manuscript.

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
