# Peer review of "Mapping liquid water content in snow at the millimeter scale: An intercomparison of mixed-phase optical property models using hyperspectral imaging and in situ measurements"

_The Cryosphere, 2021_

## Author Comment (AC1)

**Authors' reply to referee comments on "Mapping liquid water content in snow: An intercomparison of mixed-phase optical property models using hyperspectral imaging and in situ measurements" (Manuscript tc-2021-247)**

We thank the referees for taking the time to evaluate our manuscript and provide valuable feedback that will improve this manuscript. In the following, we reply to the comments of both referees and outline how we will incorporate their comments into the revised manuscript. For clarity, referee comments are in black text and the authors response is in blue text. All figures and tables that have been updated are included at the end of this document.

**Referee #1: Ryan Webb**

Comment on tc-2021-247 Ryan Webb (Referee)

This study uses NIR spectral reflectance measurements to estimate the LWC of snow, and compare those estimates to an independent dielectric measurement. This investigation conducts laboratory scale tests to determine which reflectance model is most appropriate. This model is then applied in the field to a single snow pit face to demonstrate the field applicability. This study was able to show a final uncertainty of ~1% for larger grain sizes as methods were not very promising for small grain sizes (not surprising, but excellent effort that was worth a shot).

Overall, I really liked reading this paper. It is well written and easy to follow. These methods will advance capabilities in observing LWC in the snow, a long-standing challenge in the field. With that being said, there are a few improvements that I think could really help expand the impact of the final paper. Most of these are relatively minor (at least I think so) and should not be too difficult to address. I put the comments in order of appearance in the paper, generally listed by line number. One general comment is that there are a number of recent studies investigating water flow through snow that could/should be cited (some are listed in the comments below).

Thank you Ryan, we were glad to hear you liked the paper, and we really appreciate your thoughtful comments based on your expertise on this topic.

Title: given the recent advances in mapping liquid water content at various scales (from pore scale to hillslope and watershed scales), I think specifying what scale this paper addresses in the title is justified.

The title has been updated to include the scale and is now: *Mapping liquid water content in snow at the millimeter scale: An intercomparison of mixed-phase optical property models using hyperspectral imaging and in situ measurements*

L 23: "unprecedented detail" may be a bit of an overstatement. I think the method has the potential to do so in the future, but not in the current study. The previous paper that comes to mind is Williams et al. (2010) who were able to provide a 3-D model of a 1 m x 1 m cube of snow at 1 cm^3 resolution showing the meltwater flow patterns throughout. While the current study has higher resolution, it is limited to a single pit face. However, the Williams study should be referenced and potentially added as a comparison in the discussion.

Williams, M.W., Erickson, T.A. and Petrzelka, J.L. (2010), Visualizing meltwater flow through snow at the centimetre-to-metre scale using a snow guillotine. Hydrol. Process., 24: 2098-2110. https://doi.org/10.1002/hyp.7630

The method presented by Williams et al. (2010) provides a 3-D model of meltwater at the snowpit scale which is used to quantify the spatial distribution of meltwater. We agree that it does provide more detail in terms of quantifying pooling water and flow feature distribution within a snowpack, however it does not provide a LWC measurement. Although the NIR-HSI method presented here is only 2-D, it does map measured LWC at the millimeter scale, which has yet to be done. To be more specific about what is "unprecedented" the last sentence in the abstract has been changed to read:

*Furthermore, the LWC retrieval method was demonstrated in the field by imaging a snowpit sidewall during melt conditions and mapping LWC distribution in unprecedented detail, allowing for visualization of pooling water and flow features.*

We have also included the Williams et al. (2010) reference, see the below comment for more details.

L66-70: Please clarify a little more as these processes often happen at the same time in a natural snowpack. Especially early in the melt season (Hirashima et al., 2019; Eiriksson et al., 2013). These studies could also improve the discussion as to potential future applications of the presented methods.

 Hirashima, H., Avanzi, F., & Wever, N. (2019). Wet-snow metamorphism drives the transition from preferential to matrix flow in snow. Geophysical Research Letters, 46, 14548– 14557. https://doi.org/10.1029/2019GL084152

Eiriksson, D., Whitson, M., Luce, C.H., Marshall, H.P., Bradford, J., Benner, S.G., Black, T., Hetrick, H. and McNamara, J.P. (2013), An evaluation of the hydrologic relevance of lateral flow in snow at hillslope and catchment scales. Hydrol. Process., 27: 640-654. https://doi.org/10.1002/hyp.9666

We recognize the general discussion on liquid water in snow did not fully capture water infiltration processes in a natural snowpack. *Section 2.1 Liquid Water in Snow* has been expanded to clarify the predominate water infiltration mechanisms (i.e., matrix and preferential flow), conditions under which they are observed, and how preferential flow transitions to matrix flow during the melt season. Here, Eiriksson et al (2013) as been added as a reference and Hirashima et al. (2019) has been added as a reference in discussion section 5.4 to discuss future applications of the NIR-HSI method. This section now reads:

*Water infiltration through snow is a spatially and temporally complex process, controlled by water saturation level, snow microstructure, and topography. Generally, water infiltration is described by two primary mechanisms: homogenous matrix flow and heterogeneous preferential flow. Matrix flow is described as the semi-uniform vertical movement of water, while preferential flow are concentrated water pathways that follow the path of least resistance that can extend deep into the snowpack, ahead of the matrix flow (Schneebeli, 1995). Although gravitational forces primarily drive vertical movement of water in snow, large amounts of water can be diverted horizontally due to stratigraphic layers in the snowpack, such as ice crusts or capillary barriers (i.e., fine grains over coarse grains) (Waldner et al., 2004; Webb et al., 2021; Eiriksson et al., 2013). As the snowpack becomes less stratified throughout the melt season, the general pattern transforms from preferential flow to homogenous flow (Webb et al., 2018).*

L80: The Williams et al. (2010) study quantified spatial distribution using dye tracers.

The snow guillotine study by Williams et al. (2010) has been added to the dye tracer background discussion and now reads:

*While these methods remain primarily a qualitative visualization technique, Williams et al. (2010) quantified the three-dimensional (3D) spatial distribution of meltwater within a 1 m³ snowpack using dye tracers and serial-section imaging. The 3D data was binarized into dry and wet categories to quantify flow features at the centimeter scale, but LWC is not obtainable using this method.*

L207-209: So an entire snowpit profile was taken prior to imaging. What effect do you think this had on the liquid water content of the pit face. For example, Shea et al. (2012) found that 90% of temperatures changed in a statistically significant manner in the first 90 seconds of exposure to the air and continued to change over time. I think you still demonstrate the applicability of this method in the field, but this should be a consideration for future studies that focus on the field applications.

Shea, C., Jamieson, B., and Birkeland, K. W.: Use of a thermal imager for snow pit temperatures, The Cryosphere, 6, 287–299, https://doi.org/10.5194/tc-6-287-2012, 2012.

In the field, the face of the snowpit was shaded from the sun, but the ambient air temperature was above freezing which would increase the LWC on the snowpit wall with increasing exposure time. To minimize this effect, we cut the snowpit wall back approximately 10 cm prior to imaging. Immediately following imaging, the SLF sensor measurements were taken. While working quickly as possible, this entire process likely took ~7 minutes to complete. This step has been added to the method section.

*Prior to imaging the snowpit sidewall, the entire face of the snowpit was cut back ~10 cm to minimize impacts from exposure to ambient air temperature.*

Additionally, a new section (5.3.4) has been added to discuss the uncertainties related to exposure time and it reads:

*Acquiring a vertical profile of LWC, using any instrument, requires digging a snowpit and exposing the sidewall. The atmospheric exposure can change snow properties and introduce uncertainties to the measurement. Shea et al. (2012) observed that a statistically significant change in the surface temperature across a snowpit wall occurred within the first ninety seconds of exposure, providing evidence that atmospheric equalization affects the surface temperature more strongly than from heat behind the snowpit wall. Here, the snowpit was isothermal and the air temperature was above freezing (10 °C) meaning that any additional energy into the snowpack directly goes to melting snow and increasing LWC. To minimize these uncertainties, a systematic approach was taken where the snowpit sidewall was cutback ~10 cm prior to taking hyperspectral images and SLF sensor measurements. Additionally, the snowpit was covered with an opaque tarp blocking all diffuse sunlight on the snowpit wall. Acquiring two hyperspectral images took approximately 2 minutes while acquiring the 28 SLF sensor measurements in a vertical profile took approximately 5 minutes, resulting in the snowpit sidewall being exposed for ~7 minutes. The comparison of the two instruments cannot occur simultaneously and requires the snowpit to be exposed for an extended time, which would lead to uncertainties if the LWC on the snowpit wall increased during this time.*

L225-226: Maybe an iterative approach could overcome this? Essentially, the sum of LWC mass and dry density should equal the density cutter bulk density. For example, if the SLF sensor gives you 6% LWC, that is 60 g of water for a 1 L density cutter, and if the dry density estimate for that is 350 kg/m^3, then you can compare 410 g (350 + 60) to the actual bulk measurement from the cutter and adjust accordingly. The SLF sensor records and logs the measured permittivity with the calculated LWC so this should be relatively straight forward using the equations in the manual. Feel free to reach out if this does not make sense.

We appreciate the suggestion, and after looking back at the manual we realized it was also mentioned as a resolution for the use of wet snow density by the instrument manufacturer (FPGA Company, 2018). We implemented the iterative approach, determining the unknown uncertainties related to using a wet snow density as an input. All SLF sensor LWC measurements taken in the field have been corrected using the suggested iterative approach. The correction resulted in measured LWC values increasing by 0.4-1.2%. Subsequently, Figure 11D (formerly Figure 10D) and Table 2 have been updated and included below.

Additionally, a brief description of the iterative approach has been added to the methods section and reads:

*Following the methodology proposed by FPGA Company (2018) this error was corrected by subtracting the mass of water from the wet snow density based upon the initial LWC. The updated density was used to calculate an updated LWC using the empirical calibration equation. This calculation was repeated, with each iteration returning a smaller change in snow density. Through multiple iterations it converges on the dry snow density and an updated LWC.*

L242: I do not recall if you specify that you are referring to volumetric LWC. Please double-check that this is done somewhere.

In the method section 3.3.1 the derivation of the models states that it is a volume-weighted LWC. Additionally, a flowchart has been created and added to the method section that states that the model is using a volumetric LWC.

Figure 7: This is a really cool figure. It did take me a minute, though, to get used to blue being dry and yellow being wet. Please consider inverting the color scheme. Also, a scale bar and dashed lines for the SLF sensor footprint could be quite helpful to see how much variability occurs in the footprint of the instrument. Possibly even histograms of just the SLF sensor footprint, if they differ from the whole ROI.

The color scheme of Figure 8 (formerly Figure 7) has been inverted, although a slightly different yellow-green-blue color scheme has been used to reduce the brightness of the yellow color for visualization purposes. For consistency, all figures showing LWC measurements have been updated to use the same color scheme. A scale bar has been added inside of panel 8A. Each panel (8A-D) is a single region of interest which corresponds to the footprint of the SLF sensor. The corresponding SLF sensor measurement has been added to the panel description.

L338: did you take grain size measurements at the end of the experiments also to look at grain growth? This might also have an impact (as you mention in the discussion).

The physical grain size measurements were not taken at the end of the experiments because the focus of the initial study was to retrieve accurate LWC. The effective grain growth component of this study was added after experimentation because of the substantial difference in grain size retrieval between the two optical methods used (i.e., scaled band area versus the residual method).

L339: "this data" please correct to "these data"

This has been corrected in the revised manuscript.

L345: the 2% RMSE is also the stated accuracy for the SLF sensor I think (maybe worth mentioning), so these are great results!

From personal communication with FPGA Company, the SLF sensor does not have a stated accuracy. Additionally, there are no published studies comparing the SLF sensor to other LWC measurement techniques or instruments. Using the empirical calibration curve that was derived from adding known amounts of water to dry snow, we calculated an RMSE of ~1.2% and have reported this value in the uncertainty discussion in section 5.3.3.

L474: Please see previous comment for iterative approach.

This has been addressed, see response to previous comment.

L511: I was really hoping for a lot more discussion towards this in the 'discussion' section. Questions kept popping up like: why is this useful? How might this method be helpful in the field for future studies? What improvements can you suggest to reduce the impact of the rapid changes after exposing the face (as previously mentioned) and to capture the entire pit face down to the ground? etc. A paragraph or two to this affect could really help the impact of the paper.

A new section, *5.4 Implications for Future Applications*, has been added to the manuscript to include a longer discussion about how this method could be used in the future and it reads:

*Controls on liquid water movement through snow include LWC, grain size, and wet snow metamorphism (Hirashima et al., 2019). Multi-dimensional snowmelt models have been developed to represent the relationship between grain growth and water percolation (Hirashima et al., 2014), but limited observations for validation at high spatial scales currently exist. Being able to coincidently map $r_e$ and LWC spatial distributions in such high detail could support processes-based studies and validate models coupling wetting front propagation with grain size and grain growth.*

*Coincident $r_e$ and LWC maps could also be used to better interpret microwave remote sensing retrievals and for comparison to microwave radiative transfer models, such as the Microwave Emission Model for Layered Snowpacks (MEMLS) (Wiesmann and Mätzler, 1999) and the Dense Media Radiative Transfer Multi-Layer model (DMRT-ML) (Picard et al., 2013). Generally, these models are initialized with physical snow properties, including $r_e$, LWC, and layer thickness. Currently, discrete in situ measurements of $r_e$ and LWC measurements are taken using instruments, such as IceCube (Zuanon, 2013) and SLF sensor, respectively. These instruments are generally affordable, portable, and do not require a large snowpit working space. However, to measure snow properties multiple instruments are needed and measurements would be discrete and asynchronous, making it challenging to capture spatial variability. Additionally, multiple measurements require the snowpit wall would be exposed for longer, increasing uncertainty in the measured snow properties. Although logistically more challenging to implement in the field, the NIR-HSI method addresses these limitations by measuring $r_e$, LWC, and layer thickness in high resolution simultaneously while minimizing the time the snowpit wall is exposed to ambient air.*

*There is also broader relevance for the assessment and development of snow property retrievals from measured spectral reflectance with upcoming satellite imaging spectrometer missions. These include the Surface Biology and Geology (SBG) imaging spectrometer mission and the Copernicus Hyperspectral Imaging Mission (CHIME). Although algorithm suites have been developed to retrieve snow properties from airborne imaging spectroscopy (Painter et al., 2013), LWC is not a standard part of the retrieval, and has only been demonstrated as a case study (Green et al., 2002). Time-series mapping of LWC could be used to monitor melt initiation and how it varies with slope, aspect, and elevation.*

L515-523: This final paragraph reads much more like discussion than conclusions, in my opinion. I recommend moving to the discussion section.

This section has been moved to section *5.4 Implications for Future Applications*.

I would like to re-iterate how much I like this paper and my comments are meant to be constructive. Please feel free to reach out by email or in the interactive discussion if something that I wrote is unclear or you wish to discuss further.

-Ryan Webb

**Referee #2: Chander Shekhar**

Comment on tc-2021-247 Chander Shekhar (Referee)

The article is written well and convey properly the work carried out by the authors. The objectives were mentioned clearly by authors in lines 59-63 of paper. This study focused on quantitative two dimensional retrievals of snow surficial liquid water content (LWC) using near infrared hyperspectral imaging (NIR-HSI) measurements in two scenarios. Firstly, LWC was mapped in controlled laboratory scenario by comparing reflectance spectra (measured using NIR-HSI) of snow samples (07Nos.) at different time steps with simulated reflectance spectras of three theoretical models by tuning LWC and grain size parameters. The retrieved LWC values were compared with dielectric measurement (SLF Sensor) based LWC of same snow samples. It led to selection of best matching hyperspectral imaging based retrieval model among the three. RMSE was found to be around 1% in LWC for large retrieved snow grain size values (>176μm and <1000 μm). The LWC retrieval models did not performed well for small grain sizes with complex shapes. Secondly, in field scenario the applicability potential of selected NIR-HIS based model was demonstrated for quantitative two dimensional surficial LWC retrieval on the wall of snow pit with an artificial source of illumination. The efforts by authors are appreciated as limited work is available on quantitative two dimensional LWC using hyperspectral imaging sensors.

There are a few observations and suggestions that i think will help to improve the overall impact of the paper. Specific comments include concerns based on curiosity for actual applications and other minor comments/suggestions have been put in order of appearance in the paper, listed by the line numbers.

Thank you for taking the time to read the manuscript and provide feedback. The comments were helpful particularly in regard to field applications of this method and making the case for its future utility.

**Specific Comments**

1. For field retrievals of LWC using NIR-HSI, dry snow densities could not be measured that lead to certain uncertainties in LWC. Also, few of the potential sources of uncertainty were discussed in section 5.3 without any quantitative values for discussed ones (like spherical grain size, dry snow density, illumination source etc.). For lab scenario, what LWC would have been obtained if one proceeds without accounting for dry snow density, as happened in field scenario. The authors had already worked out for lab samples by inclusion of dry snow density values. Based on this, an uncertainty figure may emerge (for 07 samples) and it will provide confidence for actual field retrievals. An observation on initial and final snow parameters in lab experiment may also help. Authors can include discussion on this in uncertainty section (Section 5.3).

   Section 5.3 has been updated in the revised manuscript to expand on the uncertainties and now includes subsections *5.3.1 Simulated Snow Reflectance, 5.3.2 Measured Reflectance 5.3.3 Dielectric Liquid Water Content Measurements, 5.3.4. Field Measurements.*

   The LWC measurement uncertainties related to using a wet snow density have been resolved using an iterative process suggested by referee #1 and described in the SLF User Manual (FPGA Company, 2018). The measured LWC uncertainty was found to be 0.4-1.2% and this has been added to the manuscript. Subsequently, measured LWC from the field experiment has been updated in the manuscript with the corrected LWC measurement, resulting in a smaller difference between NIR-HSI and SLF sensor LWC measurements. Additionally, the accuracy of the SLF sensor (~1.2%) has been added in the discussion.

2. The envisaged potential applications areas (in field/ air/ space borne modes) using proposed high resolution NIR-HSI method for quantitative two-dimensional LWC may kindly be mentioned in introduction or background section (Line 95). It will provide clarity to readers why LWC details at high resolutions are required with advantages/ limitations over low resolution LWC. Suitable references may also be included.

*We have updated the concluding sentence of this paragraph to mention envisaged application areas, it now reads:*

*Therefore, there is currently no in situ method to effectively quantify spatial variability of LWC at a high (sub-cm) spatial resolution, which could be used, for example, to validate process-based modeling of wet snow (Hirashima et al., 2019) or initialization and validation of microwave radiative transfer models (Wiesmann and Mätzler, 1999; Picard et al., 2013)*

*In the discussion, we have also expanded upon these applications in section 5.4 Implications for Future Applications.*

3. Power source of halogen lamps may be mentioned which were used in field conditions for illumination purpose. A discussion on advantages/ disadvantages of dielectric based (e.g. SLF sensor) and NIR-HSI based sensors (in terms of outputs obtained by both instruments, performance time, cost effectiveness, uncertainties involved, portability in field, preferable method/instrument for specific application etc.) may help to provide better insights to readers.

*In the field experiment line power (120V AC) was used from the weather station, we have added this detail to the methods:*

*The snowpit wall was illuminated with two 500-watt halogen lamps mounted on a tripod, line powered through the weather station.*

*A discussion of the advantages and disadvantages of the NIR-HSI method compared to other portable LWC and specific surface area/grain size measuring instruments has been integrated into the discussion of future applications for microwave remote sensing and reads:*

*Coincident $r_e$ and LWC maps could also be used to better interpret microwave remote sensing retrievals and for comparison to microwave radiative transfer models, such as the Microwave Emission Model for Layered Snowpacks (MEMLS) (Wiesmann and Mätzler, 1999) and the Dense Media Radiative Transfer Multi-Layer model (DMRT-ML) (Picard et al., 2013). Generally, these models are initialized with physical snow properties, including $r_e$, LWC, and layer thickness. Currently, discrete in situ measurements of $r_e$ and LWC measurements are taken using instruments, such as IceCube (Zuanon, 2013) and SLF sensor, respectively. These instruments are generally affordable, portable, and do not require a large snowpit working space. However, to measure snow properties multiple instruments are needed and measurements would be discrete and asynchronous, making it challenging to capture spatial variability. Additionally, multiple measurements require the snowpit wall would be exposed for longer, increasing uncertainty. Although logistically more challenging to implement in the field, the NIR-HSI method addresses these limitations by measuring $r_e$, LWC, and layer thickness in high resolution simultaneously while minimizing the time the snowpit wall is exposed to ambient air.*

4. The parameters of actual snowpack and those measured on snow pit wall are expected to be different as exposure of snow pit wall while digging leads to changes in snow pack parameters on the wall. It would be interesting to know whether illumination using halogen lamps and opaque tarp is imperative for HSI based field measurements. In a natural scenario, the signatures from snow pit wall under shadow will be of diffused sunlight and may affect the reflectance measurements and hence retrievals based on simulated reflectances. I am curious to know about applicability potential of NIR-HSI retrieval method in actual field scenarios.

   Exposure of a snowpit wall will affect the LWC at the surface of the snowpit and the steps that we took to mitigate this have been added to the manuscript, please see response to referee #1. For a snowpit application, where diffuse light is the primary illumination source on the snowpit wall, using a known direct lighting source and opaque tarp is likely the best approach for comparison to simulated reflectance because it minimizes uncertainty due to variable illumination conditions. Using a tarp and illumination source is common practice for other optical methods applied in snowpits, such as contact spectroscopy (Painter et al., 2007; Skiles and Painter, 2017). We have added a discussion on illumination conditions in the field in Section 5.3.3 Measured Snow Reflectance:

   *There is uncertainty in the measurement of absolute spectral reflectance, the accuracy of which is important for using the residual method. To minimize this uncertainty in the laboratory, spectral measurements were taken with consistent lighting conditions and at close proximity, resulting in near perfect conditions, which was ideal for the comparison study. Similarly, uncertainties related to lighting conditions in the field experiment were minimized by blocking all sunlight with an opaque tarp and illuminating the snowpit wall with a known lighting source. This approach is recommended and used with other optical methods like contact spectroscopy (Painter et al., 2007, Skiles and Painter, 2017). Using natural light in a standard snowpit orientation (i.e. facing away from the sun) would make the conversion from radiance to reflectance challenging because the snowpit wall is unevenly illuminated by diffuse light, and would need to be accounted for in the radiative transfer modeling. Additionally, there may not be enough light on the snowpit wall for imaging, however, this was not tested here.*

5. **Refer Figure 10**: To obtain the snow pack stratigraphy, one has to dig a snow pit. Ice layers can be easily recognised visually in layers of snow. It is known fact that LWC will have high values above the impermeable ice layer for a wet snow pack and can be measured easily using portable SLF sensor in any illumination condition. Definitely, resolution of NIR-HSI is better that SLF sensor. From application point of view, it is curiosity to know where and how this high resolution details will actually help, knowing the fact that (i) LWC has high spatial and temporal variability in snow pack and (ii) hyperspectral information retrieval is constrained to surface measurements only. Kindly mention.

   A new section (*5.4 Implications for Future Applications*) has been added to the manuscript to discuss how high spatial resolution simultaneous LWC and grain size measurements could be beneficial. Please see the response to a similar comment from referee #1 where we have included the added text to the revised manuscript.

The specific comments express concern towards application potential of NIR-HSI based LWC retrieval methods with due recognition to actual field constraints, variability in field spectral reflectance signatures caused by number of dynamic factors (e.g. dynamic snow parameters, viewing/ illumination conditions and geometry etc.) and limited penetration of NIR radiations into the snow.

**Other minor comments/suggestions:**

**L 1-3**: Inclusion of appropriate scale in the title of paper at which work had been performed will provide clarity. As this work had not been tested for air/space borne sensors to cover large scales.

The title has been updated to include the scale and is now: *Mapping liquid water content in snow at the millimeter scale: An intercomparison of mixed-phase optical property models using hyperspectral imaging and in situ measurements*

**L 15**: 'determine' to be changed to 'determined'.

This has been corrected in the revised manuscript.

**L23**: 'unprecedented detail' appears to be an exaggeration. Kindly use appropriate word/details.

We agree that the NIR-HSI method does not provide unprecedented detail in terms of quantifying pooling water and flow feature distribution within a snowpack. However, the NIR-HSI method does map measured LWC at the millimeter scale, which has yet to be done, therefore validating the use of "unprecedented" in those terms. To be more specific about what is "unprecedented" the last sentence in the abstract has been changed to read: *Furthermore, the LWC retrieval method was demonstrated in the field by imaging a snowpit sidewall during melt conditions and mapping LWC distribution in unprecedented detail, allowing for visualization of pooling water and flow features.*

Additionally, please see a similar response to referee #1 above for additional detail on a comparison to Williams et al. (2010).

**L149:** Inclusion of workflow diagram in Section 3 will provide quick easy understanding to the readers about the work that was carried out.

We have created a workflow diagram and inserted this into the revised manuscript as Figure 3. The new figure is attached below.

**L 175:** In laboratory setup, the distance (m) of halogen lamps and NIR-HSI imager from snow samples may kindly be mentioned.

In the laboratory, the halogen lamps and NIR-HSI imager were 38 cm and 47 cm from the surface of the snow sample, respectively. This information has been added to the revised manuscript.

**Table 1**: A column may be added for retrieved grain size values using SBA method for dry snow samples, that corresponds to grain size legend of Figure 8.

Table 1 has been updated to include this information.

**L 205-206**: The distance (m) of halogen lamps and NIR-HSI imager from snow pit wall may be mentioned.

The distance of the halogen lamps and the hyperspectral imager from the snowpit wall has been added to the revised manuscript and now reads:

*The imager was mounted on to the Resonon Outdoor Field System, which includes a tripod mounted rotational stage, and was placed 110 cm from the snowpit wall. The snowpit wall was illuminated with two 500-watt halogen lamps mounted on a tripod, which were placed perpendicular to the wall at a distance of 90 cm, similar to the laboratory setup presented in Donahue et al. (2021).*

**L 216:** What were the implications of using 36% reflectance calibration panel during field experiments in place of 99% (that was used in lab scenario)?

The difference in the % reflectance of the calibration panels should not impact the conversion of radiance to reflectance if the difference in reflectance is correctly accounted for. The 36% reflectance panel was advantageous for the field experiment because it was large enough to perform a pixel-by-pixel calibration of the entire snowpit face, minimizing illumination imperfections. Placing the smaller 99% reflectance targets in the scene of the snowpit face would also work, but this type of calibration would not account for minor differences in the illumination across the snowpit face.

**Figure 4:** The figure may be modified to depict clearly that upper 110cm of snowpit wall was imaged using NIR-HSI. It appears as if image was taken up to 150cm depth.

The bottom 40 cm of the snowpit face is cross-hatched in black to indicate that it was not imaged and is also noted in the caption and in the text.

**L 225:** For words 'a small error', a quantified value may be mentioned.

These words have been deleted from the revised manuscript because the "small error" was corrected by using the iterative approach, which is discussed in previous comments.

**L 250–254**: It appeared that single scattering properties were tunable using only two parameters LWC and $r_e$. Are there any other parameters also which have not been considered, under some assumptions/ approximations? These can be mentioned.

In addition to ice particle size and liquid water content, light absorbing particles (LAPs, e.g., dust and black carbon) impact spectral reflectance of snow and can be accounted for in radiative transfer simulations. Here, we assume clean snow (i.e., no LAPs) which was the case for the laboratory samples. For field applications this is still a reasonable approach because at relatively low concentrations, which are commonly found in natural snow, light absorbing particles do not impact the NIR portion of the spectra. This assumption has been added to section *3.3.1 Single Scattering* and a brief discussion of the effects of light absorbing particles has been added to the discussion section and reads:

*The simulated reflectance spectra used in this study assume clean snow (i.e., no LAPs). This assumption was valid for the laboratory experiments, however snow in natural environments contain varying amounts of LAPs. Since LAPs predominately effect the visible part of the spectrum (Nolin and Dozier, 2000), it is unlikely that low concentrations of light absorbing particles will impact this retrieval method. Conversely, high concentrations of LAPs impact the NIR part of the spectrum (Skiles et al., 2018) and would likely negatively impact this retrieval method.*

**Figure 7:** SLF sensor measured LWC values corresponding to time steps in (A to D) may also be mentioned.

Figure 7(A-D) have been updated to include the corresponding SLF sensor measurements in each panel.

**L 301-304**: Criteria or reference statistical parameter (RMSE, Bias etc.) can be mentioned for the quoted relative best or poor performance of models.

The intent of the opening paragraph is to give an overview of the main results with a full justification of the findings described in the following paragraphs (Lines 345-349).

**Figure 8:** Legends are same for Figure 8 (A – C) and can be placed outside the three figures to have better representation. Author can take a decision on this based on editor's suggestions.

We prefer to leave the legends located inside of Figures A and B because it allows for the plots to be as large as possible within the margins. Moving the legends outside of the figure on the right or left side would make the plots smaller and harder to view. If the editor would prefer the legends to be outside of the figures, we could move the legends below the figure in a horizontal format, however the varying size of the scatter plot symbols may not be as obvious as in a vertical format.

**Table 2:** (1) The number of sample points used to arrive at RMSE and Bias corresponding to each sample need to be mentioned. The confidence level may also be mentioned. (2) The word 'RSME' need to be spelled correctly.

Table 2 has been updated to include the number of data points (n) for each sample. We have decided to forego including a confidence level in this table because we are reporting the RMSE compared to the SLF sensor and a confidence level does not add value to this analysis. The word "RSME" has been corrected to "RMSE" in the revised manuscript.

**Caption of Figure 10:** The words 'depth average of the SLF sensor area only' may be modified to provide clarity that it is HSI measurement corresponding to area covered by SLF sensor only. Description of 'NaN' can also be mentioned.

The caption of Figure 11 (formerly Figure 10) has been updated to provide clarity that the "Depth Avg" and the "Sensor Area Depth Avg" liquid water content profiles are from the hyperspectral imager (NIR-HSI). Additionally, NaN has been defined and a short description mentioned. The caption now reads:

*Figure 11: Results from snowpit at Bridger Bowl Ski Area. (A) Density profile using 1000 cc wedge cutter and depth averaged effective grain radius. (B) Map of effective grain radius across snowpit wall. (C) Map of liquid water content across snowpit wall with the SLF sensor vertical profile area outlined in dashed red line (D) Liquid water content profiles from the SLF Snow Sensor, NIR-HSI depth averaged across the entire width of the snowpit, and NIR-HSI depth averaged over the area covered by the SLF sensor area only. NaN (not a number) values in B and C correspond to no retreival data due to a ruler that was placed in the scence.*

**Table 3:** Spelling corrections: 'Senor' to be replaced by 'Sensor'.

This has been corrected in the revised manuscript.

**L 454-457:** Kindly check whether these lines convey the correct message. It seems like inclusion of certain additional points has led to smaller SBA. Kindly verify.

We have verified the correct message is conveyed in these lines, we do not include additional points (bands) in the retrieval. In Figure 12C (formerly 11C) we are highlighting the change in the absorption feature shape due to the presence of water, which results in a smaller scaled area. This is attributed to the

scaled band area method having fixed endpoints which cannot capture the shift of the feature to shorter wavelengths, resulting in a scaled band area bias.

**L 469**: A quantitative figure or reference for uncertainty of SLF sensor may be mentioned.

We have updated the uncertainty discussion (section 5.3.3) to include the RMSE of the SLF sensor, which is 1.2%.

**L 515**: 'un-precedented detail' appears to be an exaggeration. Kindly use appropriate word/ details.

To our knowledge, no other method can map effective grain size and LWC at high (millimeter scale) spatial resolution which would warrant the use of the words "unprecedented". Please see above comments for a more detailed response.

**L 519-524**: While using HSI based retrieval methods, kindly suggest the ways to account for the expected variability in reflectance signatures caused by illumination changes for air/space borne sensors. In this air/space borne sensor scenario, the level of uncertainty expected in LWC mapping using proposed simulated spectra approach may also be commented.

Postulating a level of uncertainty for application of this method to airborne or spaceborne sensors is beyond the scope of this study. We do address illumination conditions for field conditions, though, as discussed in response to specific comment #4 above.

The article presented analysis on LWC retrieved from spectral reflectance signatures (in image form) and LWC measured using dielectric based, point form data acquired in synchronization for fair comparisons. It is nice set of information that will help research community to understand the potential of hyperspectral data for retrieval of LWC parameter of snow. The comments/ observations have been written with a constructive and curious spirit to improve the impact of the paper.

Best wishes.

Chander Shekhar

**Updated Figures**

[Figure]

**Figure 1: Liquid water content and effective grain radius retrieval and assessment workflow**

[Figure]

**Figure 2: (A-D) Time series liquid water content mapping over a single region of interest during a laboratory experiment. (E) Liquid water content distribution in images (A-D) shown in a histogram.**

[Figure]

**Figure 3: Mean effective grain radius ($r_e$) retreival comparison between residual and scaled band area methods using the interstitial sphere model at each time step for Samples 1-7. Best fit line is drawn through the dry snow condition for each sample.**

[Figure]

**Figure 4: Results from snowpit at Bridger Bowl Ski Area. (A) Density profile using 1000 cc wedge cutter and depth averaged effective grain radius. (B) Map of effective grain radius across snowpit wall. (C) Map of liquid water content across snowpit wall with the SLF sensor vertical profile area outlined in dashed red line (D) Liquid water content profiles from the SLF Snow Sensor, NIR-HSI depth averaged across the entire width of the snowpit, and NIR-HSI depth averaged over the area covered by the SLF sensor area only. NaN (Not a Number) values in B and C correspond to no retreival data due to a ruler that was placed in the scence.**

**Updated Tables**

**Table 1: Laboratory Snow Samples (reported $r_e$ is from scaled band area method).**

| Sample Number | Description | Sieve Size (mm) | Initial $r_e$ (μm) | Dry Density (kg/m³) | Warming Time (min) |
|---|---|---|---|---|---|
| 1 | PP, Snowmaker Snow | 2 | 113 | 115 | 111 |
| 2 | DF, Snowmaker Snow | 2 | 130 | 212 | 161 |
| 3 | RG, Natural Snow | 2 | 176 | 455 | 118 |
| 4 | MF, Natural Snow | 5 | 398 | 440 | 103 |
| 5 | FC, Natural Snow | 2.5 | 463 | 493 | 86 |
| 6 | MF, Sample 4 Melt/Refreeze 1x | N/A | 699 | 468 | 99 |
| 7 | MF, Sample 4 Melt/Refreeze 2x | N/A | 898 | 510 | 208 |

**Table 2: Liquid water content retrieval results from the laboratory (reported $r_e$ is from scaled band area method).**

| Sample Number | n | Initial $r_e$ (μm) | %LWC Range | RMSE (% LWC) $k_{eff}$ Spheres | RMSE (% LWC) Coated Spheres | RMSE (% LWC) Interstitial Spheres | Bias (% LWC) $k_{eff}$ Spheres | Bias (% LWC) Coated Spheres | Bias (% LWC) Interstitial Spheres |
|---|---|---|---|---|---|---|---|---|---|
| 1 | 66 | 113 | 0 - 14.6 | **8.7** | 11.9 | 9.4 | **7.5** | 10.5 | 8.0 |
| 2 | 102 | 130 | 0 - 15.9 | **7.7** | 11.5 | 8.6 | **7.1** | 10.8 | 7.9 |
| 3 | 78 | 176 | 0 - 17.2 | **2.0** | 7.0 | 2.6 | **1.0** | 5.8 | 1.6 |
| 4 | 123 | 398 | 0 - 13.4 | 1.8 | 4.0 | **0.9** | -1.5 | 3.2 | **-0.5** |
| 5 | 132 | 463 | 0 - 15.2 | 1.9 | 4.7 | **1.1** | -1.5 | 4.0 | **-0.3** |
| 6 | 102 | 699 | 0 - 8.8 | 1.6 | 2.8 | **1.0** | -1.2 | 2.3 | **0.3** |
| 7 | 90 | 898 | 0 - 10.4 | 2.2 | 3.3 | **1.4** | -1.6 | 2.3 | **-0.1** |

**Table 3: Liquid water content retrieval results from the field.**

| | SLF Snow Sensor | $K_{eff}$ Spheres Sensor Area | $K_{eff}$ Spheres Full Profile | Coated Spheres Sensor Area | Coated Spheres Full Profile | Interstitial Spheres Sensor Area | Interstitial Spheres Full Profile |
|---|---|---|---|---|---|---|---|
| **Mean LWC [%]** | 12.6 | 15.6 | 15.1 | 23.5 | 23.2 | 16.3 | 15.7 |
| **σ** | 2.3 | 3.2 | 2.7 | 1.8 | 1.1 | 3.4 | 3.2 |
| **n** | 28 | 21,758 | 243,386 | 21,758 | 243,386 | 21,758 | 243,386 |

**References**

[revised manuscript text omitted]